# Attention-Deficit Hyperactivity Disorder (ADHD): A Comprehensive Overview of the Mechanistic Insights from Human Studies to Animal Models

**DOI:** 10.3390/cells14171367

**Published:** 2025-09-02

**Authors:** Matthew William Yacoub, Sophia Rose Smith, Badra Abbas, Fahad Iqbal, Cham Maher Othman Jazieh, Nada Saed Homod Al Shaer, Collin Chill-Fone Luk, Naweed Imam Syed

**Affiliations:** 1Hotchkiss Brain Institute, University of Calgary, Calgary, AB T2N 4N1, Canada; matthew.yacoub@ucalgary.ca (M.W.Y.); badra.abbas@ucalgary.ca (B.A.); fahad.iqbal@ucalgary.ca (F.I.); 2Alberta Children’s Hospital Research Institute, Cumming School of Medicine, University of Calgary, 3330-Hospital Drive NW, Calgary, AB T2N 4N1, Canada; 3Department of Cell Biology and Anatomy, University of Calgary, Calgary, AB T2N 4N1, Canada; 4O’Brien Office of the Health Sciences, University of Calgary, Calgary, AB T2N 4N1, Canada; sophia.smith1@ucalgary.ca; 5College of Medicine, Alfaisal University, Ridah 11533, Saudi Arabia; cjazieh@alfaisal.edu (C.M.O.J.); nalshaer@alfaisal.edu (N.S.H.A.S.); 6Department of Clinical Neurosciences, University of Calgary, Calgary, AB T2N 4N1, Canada; ccfluk@ucalgary.ca

**Keywords:** attention-deficit/hyperactivity disorder, psychostimulants, neurodevelopment, monoamine signaling, cognitive outcomes, in utero exposure, animal models

## Abstract

Attention-deficit/hyperactivity disorder (ADHD) is a neurodevelopmental condition marked by persistent inattention, hyperactivity, and impulsivity. Despite its considerable global prevalence, key gaps remain in our understanding of the structural and molecular changes underlying ADHD which complicate adult diagnosis, as symptoms present differently from those observed during childhood ADHD. On the other hand, while psychostimulants effectively mitigate some symptoms, significant controversy surrounds their long-term effects on cognition, learning, and memory, and day-to-day living. Moreover, our understanding of how various medications given to alleviate ADHD symptoms during pregnancy impact the developing fetal brain also remains largely unexplored. Here, we discuss the subtle differences between ADHD in children and adults and how these symptoms alter brain development and maturation. We further examine changes in monoamine signaling in ADHD and how psychostimulant and non-pharmacological treatments modulate these neural networks. We evaluate and discuss findings as they pertain to the long-term use of ADHD medications, including in utero exposure, on cognitive outcomes, and contextualize these findings with mechanistic insights from animal models.

## 1. Introduction

The human brain undergoes remarkable changes from infancy through adolescence, shaping our cognitive, emotional, and behavioral traits. Neurodevelopment is a precisely timed and intricately orchestrated process, where perturbations and miscues in the formation of neural circuits can lead to neurodevelopmental disorders such as autism spectrum disorder (ASD) and attention-deficit/hyperactivity disorder (ADHD), which are often blanketed as “neurodivergence” in cognition and behavior [1]. ADHD, which affects approximately 5% of children [2] and 6.76% of adults [3], is diagnosed about twice as frequently in males as compared to females [4]. These higher numbers are likely attributable to behavioral symptoms of ADHD being more easily recognizable in males than females [5]. ADHD is generally characterized by inattentiveness, hyperactivity, and impulsivity [2]. The Diagnostic and Statistical Manual of Mental Disorders 5 (DSM-5) [6] defines the diagnosis of ADHD for children as the presence of at least 6 symptoms, including inattentiveness, hyperactivity, and impulsivity, whereas in adults this decreases to five such indicators [7]. A fundamental question that remains unanswered is as to the underlying structural and functional anomalies in the brain that are likely responsible for various observed behavioral changes in ADHD. These alterations in the maturation of brain networks, notwithstanding synaptic plasticity, may underlie diverse behavioral traits, though the precise underlying causes of ADHD and other neurodevelopmental disorders, may they be genetic, cellular, molecular, or environmental, remain largely unknown. Moreover, various behavioral therapies and medications used to treat ADHD symptoms also remain controversial.

This review will first compare ADHD symptoms in children and adults, and how they relate to brain development during childhood and adolescence. Next, we will look at how neurotransmitters contribute to ADHD symptoms, and how stimulants and non-stimulants work to attenuate them. We will then discuss the mechanisms underlying the efficacy of ADHD medications and how they impact long-term cognitive function in both humans and preclinical models. Subsequently, we will review the implications of ADHD medications when taken during pregnancy, and the use of transcranial brain stimulation as a possible future treatment. Lastly, we will finish by proposing a potential paradigm shift which could involve integrating synaptic-level findings from animal models and linking them to more complex neuronal networks in humans to better understand the underlying causes of ADHD, and the impact of psychotherapies on long-term memory.

## 2. Diagnosing ADHD in Children and Adults: Similarities and Differences

The DSM-5 [6] considers the age-of-onset limit for ADHD in children to be 12 years, whereas adult ADHD with this reduced symptom threshold is considered for ages 17 and older. In children, ADHD symptoms are often more easily discernible, particularly those within the hyperactive domain. For example, a child with ADHD may fidget during class and struggle to remain seated, or miss social cues during conversations, such as frequently interrupting others [8]. Inattentiveness in children is typically observed with difficulties in planning and completing coursework, and consistently remaining unengaged during classroom lectures, often leading to poor academic performance [9]. In adults, on the other hand, the hyperactive symptoms are often more subtle, transitioning from physical hyperactivity to more of an internal or emotional restlessness [10]. Due to these challenges, adult ADHD tends to be more challenging to diagnose, particularly because visible ADHD hyperactivity symptoms often decrease with age or are compensated for by cognitive overactivity, which can manifest itself in intrusive thoughts [11]. Thus, inattentiveness becomes the more persistent symptom of adult ADHD, which can lead to frequent employment changes, career instability, and challenges maintaining relationships [12].

The inattentiveness in adult ADHD patients appears to be more pronounced when performing complex tasks as compared to the simpler ones [13], which may limit performance in academic [14] and workplace settings [15]. Consequently, individuals diagnosed with adult ADHD often report a sense of relief from finally having an explanation provided for the challenges that they have been experiencing [16,17]. It is worth noting that ADHD does not necessarily perturb all aspects of cognitive function or behavior. For example, there has been a demonstrated link between ADHD and increased creativity and “divergent” thinking, as defined by flexibility or originality when coming up with solutions for tasks at hand [18,19]. In fact, evidence suggests that ADHD medications may impair creativity in children [20,21], which has otherwise been attributed to creative storytelling [22], and in some cases, greater achievement in artistic domains, such as music, writing, and theater [23]. Therefore, it is important to recognize that not all adults view their ADHD diagnosis as a limitation; rather, some believe that it provides them with advantages from a neurodivergent perspective [16,24]. It however remains to be seen whether this augmentation in the above traits and achievements comes at the expense of other mental faculties.

The extent of inattention and hyperactivity symptoms can vary distinctly in both children and adults; thus, ADHD is classified into subtypes of primarily inattentive (ADHD-I), primarily hyperactive (ADHD-H), or combined (ADHD-C) based on the criteria provided in Table 1.

Notwithstanding these established benchmarks, there are concerns regarding both the over- and under-diagnosis of ADHD, potentially arising from the fact that the criteria used are largely qualitative. Adding to the challenge, while ADHD itself is chronic, the symptoms in children often change in their presentation over time [32]. For adults with undiagnosed ADHD, it is possible that their symptoms may initially have gone unnoticed due to having greater structure to their day-to-day lives, alongside social support during childhood. Moreover, as compared to children, undiagnosed adults may have better adapted to their circumstances by trying to fit into social norms around them [33], thus masking their ADHD symptoms from being readily apparent. Additionally, several ADHD symptoms may exhibit overlap with ASD, such as impaired response inhibition, challenges with sustained attention, and decreased social functioning [34,35]. One study provided evidence that the largest distinguishable characteristic between ASD and ADHD is the notion of social reciprocity, where ASD children struggle more with demonstrating both social enjoyment and guiding or directing attention during conversations [36]. Additionally, in this study [36], ADHD children appeared to have externalized their problems through impulsive behavior and hyperactivity, whereas ASD children were more likely to withdraw from socialization. Therefore, while ADHD in children is comorbid with ASD in 9.8% of cases [37], ADHD does present unique symptoms that allow it to be distinguished from other neurodevelopmental disorders. Even on a day-to-day basis, the specific scenario that an individual with ADHD may experience can both improve or worsen their symptoms [38]. Thus, early childhood experiences are thought to be crucial for ADHD management, as social and educational support can lessen the progression of symptoms into adulthood which occurs in around 60% of cases [39,40,41].

As a case in point, support from parents and friends, participation in community traditions, and a sense of belonging during adolescence- all linked to greater emotional regulation for adults with ADHD [42]. Parental mental well-being, alongside using appropriate discipline strategies with their children, appears to be the best predictors of ADHD symptom persistence [43,44]. Additionally, it appears that for children with ADHD, having a caregiver whom they feel they can confide in may be a strong predictor of better self-esteem and emotional well-being [45]. For both school-aged children and adolescents, social support from friends, parents, and teachers can act as a protective buffer against stress and improve self-perception [46,47].

On the other hand, adverse childhood experiences such as socioeconomic hardship, familial mental illness, and domestic violence are linked to more severe ADHD symptoms in adulthood [43,48]. Even when no significant differences in neurocognitive functioning are present, these adults with more adverse childhood experiences report greater perceived ADHD symptoms [48]. This reinforces the importance of positive early life experiences in shaping a child’s long-term perception of their ADHD, as well as the nature of their symptom progression as their brain continues to develop.

## 3. Brain Structural Changes Associated with ADHD

During critical periods of development in early childhood and adolescence, several structural differences have been observed in the ADHD brain. For instance, although a typical pattern of development is generally seen, there are temporal delays observed as early as five years after birth [49,50]. Specifically, children with ADHD have delayed prefrontal cortex (PFC) development of approximately 3 years, reaching 50% peak cortical thickness at around 10.4 years as compared to 7.5 years in typically developing individuals (TDIs) [50,51]. This delay in PFC development is linked to an overall reduction in brain volume in this region [52]. The PFC plays an integral role in planning, attention, and decision-making, which are likely impaired in ADHD, and receives more robust projections from the nucleus accumbens (NAc) [53], the reward center of the brain. Thus, this increased signaling from the NAc likely contributes to the impulsive symptoms seen in ADHD, with reward-based decision-making exerting greater influence over cognition and decision-making [53]. It has also been demonstrated that reduced connectivity from the NAc to the right paracingulate gyrus, which is involved in cognition and emotional processing [54], correlates with more severe ADHD symptoms during both childhood and adulthood [55]. On the other hand, reduced connectivity from the NAc to the right insular cortex, which is involved in broad functions including risk-reward behavior and sensorimotor processing [56], predicts better outcomes for ADHD symptoms [55]. This suggests that an imbalance within reward circuits between the NAc and its downstream connections likely plays an integral role in the expression of ADHD symptoms. Altogether, increased reward-related signaling to the PFC in ADHD may overwhelm the brain’s capacity for self-regulation and result in impulsive behavior. Unlike TDI, children with ADHD do not see an increase in middle and right PFC activity between ages 8 and 11 [57]. However, they do see an improvement in left PFC activity during this time which does not occur in TDIs [57]. Adolescents with ADHD have shown increased blood flow to this brain region during working memory tasks which did not occur in TDIs [58]. This increased activity of the left PFC may thus be a compensatory mechanism for the delayed maturation of the other PFC regions.

A similar developmental delay has been observed in the cerebellum, which reached 50% cortical maturation at 10.5 years in children with ADHD, as compared to 7.5 years in TDIs [50]. The cerebellum plays a key role in regulating movement, balance, and coordination, alongside contributing to regulating attention, executive function, memory, language, and visuospatial abilities, and is implicated in ADHD symptoms pertaining to impaired motor control [59]. Loss of volume in the cerebellum has been demonstrated in the ADHD brain, which continues during adolescence and is associated with more severe ADHD symptoms [60,61]. ADHD symptoms regarding reduced coordination are also thought to arise from decreased volume of the corpus callosum (CC) which is observable in children [62,63]. The CC is responsible for interhemispheric communication, coordinating motor function across both sides of the body, and integrating complex cognitive processes such as problem-solving, language, and memory [64]. However, upon reaching adulthood, it appears that these reductions in CC size normalize to those of TDIs [65]. In males with ADHD around 70 years of age, inattention and hyperactivity were associated with decreased CC thickness, whereas age-matched females demonstrated a thicker CC, which was linked to increased hyperactivity, suggesting the possibility that there may likely be sex differences in the progression of hyperactivity into later stages of life [66]. It however remains unknown whether these ADHD symptoms in the elderly corroborate with age-related changes in various brain regions as observed during dementia and Alzheimer’s disease. Thus, it might be the case that there are likely sex-specific differences in CC thickness that may occur later in life for adults with ADHD. This would however be difficult to delineate due to progressive, age-related neurodegeneration which is not confined to just one brain region. The basal ganglia (BG) is another region of the brain where a loss of volume is thought to be associated with ADHD symptoms of hyperactivity, such as fidgeting [67,68,69]. Children with ADHD have a decreased surface area in the ventral striatal region of the BG from ages 8 to 18, atypical of the progressive increase in surface area seen in TDIs [68]. This region of the BG is involved in reward-based learning and decision-making [70], and thus loss of volume in this region may contribute to impulsivity in ADHD. This notion reinforces the narrative that the symptoms of ADHD are not only the result of delayed PFC development, rather there are widespread changes that occur across multiple brain regions including the frontal, temporal, parietal, cerebellar, and limbic structures.

In general, it has been demonstrated that there is an overall decrease in total brain volume in ADHD [71,72]. This occurs predominantly through a loss of white matter volume (10.7%), alongside a decrease in grey matter volume (3.9%), leading to an approximately 3% total decrease in brain volume [73].

The brain structural changes associated with ADHD are summarized in Figure 1.

Structural anomalies aside, all brain functions rely upon synaptic connectivity between the neurons, which often invoke either ion channel functions or neurotransmitter-receptor interactions. As such, a great deal of focus has been placed on fine-tuning these neuronal pathways and modulating synaptic connectivity through either the use of chemical compounds or more novel therapies like transcranial magnetic stimulation.

## 4. Neurotransmitters and ADHD

Neurons communicate via neurotransmitters—both classical (dopamine, serotonin, acetylcholine, glutamate, GABA, etc.) or peptides (norepinephrine, epinephrine, substance P, etc.) released at the synaptic cleft or neuro-hormonally. Reduced levels of dopamine (DA) and norepinephrine (NE) have historically been hypothesized to be a major cause of ADHD [74]. Evidence supporting this hypothesis stems from the fact that psychostimulants, which are the first-line treatment for ADHD, increase levels of DA and NE, and as such are deemed effective treatment of ADHD symptoms [75]. These ADHD medications are classified as either stimulants, which include amphetamine (AMPH), or methylphenidate (MPH)-based pharmacotherapies, or non-stimulants (i.e., atomoxetine (ATX)) which do not directly increase DA release. Stimulant medications have common side effects, including decreased appetite, insomnia, irritability, and increased blood pressure resulting from DA and NE release. However, stimulant medications like Adderall and Ritalin are often abused as cognitive enhancers in academic settings [76]. Additionally, the long-term effects of these medications on cognitive processes are not fully determined [77], and their long-term effects on the fetal brain during pregnancy are also not fully understood [78]. Here, we review the monoamine networks thought to be involved in the symptoms presented in ADHD.

Both stimulant and non-stimulant ADHD medications modulate the levels of monoamines that are available within the brain. This includes DA, NE, and serotonin (5-HT), which all form distinct pathways in the central nervous system (CNS). Monoamines have a fundamental role in modulating mood, cognition, and behavior [79], and perturbations in their transmission have been implicated in ADHD and the above-mentioned symptoms [80]. Despite the historical focus on these transmitters in ADHD pathophysiology, the variation in ADHD phenotypes alongside case-by-case differences in brain region-specific transmitters has made it difficult to deduce whether it is a reduction or an augmentation of monoaminergic transmission in the brain that causes ADHD symptoms.

### 4.1. Dopaminergic Signaling in the Brain

DA is involved in linking learning and reward-based decision-making [81] and reinforcement of repeated behavior [82]. At the heart of the DA system is the ventral tegmental area (VTA), which sends projections to form two key pathways (Figure 2A). The mesocortical pathway connects the VTA to the PFC, in which DA signaling plays a key role in emotional and behavioral regulation, alongside working memory and decision-making [83]. Perturbed DA levels in the PFC are thought to contribute to deficits in habit formation and planning associated with ADHD [74]. The mesolimbic pathway contains projections from the VTA to limbic structures such as the hippocampus, amygdala, and NAc. Impaired DA signaling within these networks likely underlies impulsive decision-making observed in ADHD [84]. The nigrostriatal pathway bridges connectivity between the substantia nigra (SN) and the dorsal striatum (DS), and functions to regulate control of voluntary movement [85]. Lastly, the tuberoinfundibular pathway links the hypothalamus and pituitary gland, where dopamine can influence prolactin release [86]. Thus, the dopaminergic system may serve myriad functions linking various brain regions to influence attention, movement, hormones, and long-term decision-making.

DA receptors are G-protein-coupled receptors (GPCRs), and contain 5 subtypes (D1–D5), which as classified as either D1-like (D1 and D5), or D2-like (D2, D3, D4) receptors [87]. D1-like receptors are stimulatory, where DA binding causes an elevation of cyclic adenosine monophosphate (cAMP) levels, leading to increased activity of protein kinase A (PKA). Activation of PKA leads to phosphorylation of the transcription factor cAMP response element binding protein (CREB) and molecular pathways linked to enhanced synaptic plasticity. On the other hand, D2-like receptors are inhibitory; a majority is found on non-dopaminergic neurons, whereas D2-like auto-receptors are found in dopaminergic neurons in the VTA and substantia nigra pars compacta (SNc) [115]. Activation of these receptors serves to inhibit cAMP levels and thus PKA activation, while also activating inward rectifying potassium channels (IRK) to decrease the likelihood of depolarization from occurring [116].

The historical view of ADHD as a disorder arising from hypodopaminergic signaling has been challenged in the past decade (as covered by MacDonald et al. [74] in a recent review). In humans, positron emission tomography (PET) scans can be used in conjunction with radioligands to assess the activity of dopamine transporter (DAT), DA receptors, and norepinephrine transporter (NET). Findings from Forssberg et al. [117] provided evidence for a reduced rate of DA synthesis in the midbrain of male adolescents with ADHD. This aligned well with previous studies that showed increased DAT activity in both children and adults within the striatum and basal ganglia [118,119]. A follow-up study by Ludolph et al. [120] confirmed the presence of reduced DA synthesis in adults with ADHD. While Ernst et al. [121] provided evidence that children with ADHD had 48% greater accumulation of 3,4-dihydrophenylalanine (DOPA) in the right midbrain, the data provided was however not statistically significant across multiple comparisons.

Despite these findings suggesting that hypodopaminergic signaling underlies ADHD, these findings are not consistently observed in diverse cohorts of ADHD patients. Ample evidence in the past decade has accumulated to support the notion that hypo-dopaminergic signaling is not a homogeneous hallmark of ADHD as it has been purported to be. Jucaite et al. [122] have demonstrated reduced availability of DAT in the brains of children with ADHD. These findings have been validated by Volkow et al. [123], showing greater DAT binding in the NAc, midbrain, and left caudate. Their research team subsequently correlated increased DAT activity in the NAc and midbrain with increased motivation for adults with ADHD [124]. These same ADHD patients exhibited reduced DAT availability in the very same brain regions. Similar findings have been reported by Itagaki et al. [125] for reduced DAT availability in the NAc for ADHD adults who were not exposed to ADHD psychotherapies.

While the above studies may seem contradictory, a meta-analysis by Weber et al. [126] concluded that DA followed an inverted U-shaped relationship with working memory. That is to say, both hypodopaminergic and hyperdopaminergic signaling can impair cognitive processes. Therefore, when considering data obtained from PET scans in humans, it is important to consider DA transmission through the lens of functional signaling. For example, it has been shown that there are reduced D2 and D3 receptors in the midbrain and NAc for those with ADHD as compared to TDIs [123,124]. Given that these receptors inhibit DA transmission, one interpretation could be that their lower levels may result in increased DA levels in the ADHD brain. However, it is also possible that the downregulation of these receptors might be downstream of reduced DA signaling to begin with. Therefore, at this moment, we can conclude that DA is likely involved in the pathophysiology of ADHD, and that there are synaptic-level changes that occur in the ADHD brain. It also appears that it is most likely that ADHD may be linked to both excessive and/or perturbed DA signaling, which might help explain why ADHD medications do not work for all patients. These discrepancies regarding dopaminergic signaling and its involvement may arise from varying metabolic pathways that may be active in a diverse population of ADHD patients. As shown by Ludolph et al. [120], treatment history also appears to cause intragroup variability in DA metabolism for ADHD patients; hence, it is unlikely that dysfunction in the DA system alone would be responsible for ADHD symptoms.

### 4.2. Noradrenergic Signaling in the Brain

DA metabolism involves the initial conversion of L-tyrosine into DOPA through tyrosine hydroxylase [87] (Figure 2B). Conversion of DOPA into DA is completed by aromatic L-amino acid decarboxylase and can then be converted into NE through dopamine β-hydroxylase (DBH). DBH is expressed inside the vesicles of norepinephrinergic neurons within the CNS [88]. The locus coeruleus (LC) is thought to be the primary site of NE synthesis in the brain, and innervates throughout the cerebral cortex, amygdala, cerebellum, and hippocampus [89] (Figure 2C). Perturbed NE signaling within the PFC contributes to poor concentration associated with ADHD. Additionally, NE signaling between the LC and CA3 region of the hippocampus is crucial for memory formation during new experiences [127]. NE is thought to enhance the long-term memory of emotional experiences through innervations between the LC, hippocampus, and amygdala [128]. Thus, impaired NE signaling in ADHD may also contribute to deficits in emotional memory, albeit exact mechanisms remain unclear.

Alongside DA, NE has also been historically implicated in the pathophysiology of ADHD. Until the late 2000s, methodology for assessing NET in the human brain using PET imaging was challenging due to the absence of effective radioligands [129]. Upon its availability, however, Hannestad et al. [130] reported the first in vivo findings in humans that demonstrated MPH binding to NET reduced the availability of this transporter in the brain. This verified that ADHD medications like MPH not only interact with DA signaling but also increase NE transmission. Vanicek et al. [131] assessed NET availability using PET imaging in adults with ADHD who had not been exposed to psychotherapies. They did not find differences in NET availability or distribution in the hippocampus, putamen, pallidum, thalamus, or midbrain, but did find a negative correlation between age and NET availability in the thalamus, midbrain, and LC. However, they were not able to assess cortical regions using their PET setup, where the PFC is one of the main regions exhibiting reduced NE and is believed to contribute to ADHD symptoms. Ulke et al. [132], on the other hand, found reduced NET availability in the fronto-parietal-thalamic-cerebellar regions in adults with ADHD. Further evidence to suggest impaired NET function in ADHD was provided by Singurdardottir et al. [133], where methylation of NET was negatively correlated with the severity of hyperactivity-impulse symptoms in ADHD adults. This extended their previous findings which linked mutations in the NET *SLC6A2* gene with reduced NET availability in the cerebellum [134]. Shang et al. [135] explored two specific mutations within the NET gene *SLC6A2*: rs36011 (T) and rs1566652 (G), which make up the TG haplotype. They found that children with this haplotype had reduced activity in brain regions responsible for visual memory and attention, and higher activity in regions linked to sensorimotor attention. Findings from Hohmann et al. [136] demonstrated an association between NET variants and ADHD diagnosis, and altogether suggest that genetic alterations of NET are likely implicated in ADHD.

The link between NE and ADHD was further established by Barkley et al. [137], linking polymorphisms in the *DBH* gene to increased hyperactivity in childhood, and a greater likelihood of behavioral problems in adolescence. This aligned well with previous findings of Smith et al. [138] which demonstrated a significant association between the TaqI A polymorphism in the *DBH* gene and ADHD using the same dataset. Kieling et al. [139] also provided evidence for polymorphisms within the promoter for the *DBH* gene corresponding to decreased performance in cognitive assessments for children with ADHD. These polymorphisms were also linked by Bellgrove et al. [140] to deficits in visual attention for ADHD children and adolescents. Altogether, this provides evidence for alterations in NE metabolism implicating it in ADHD; however it is still unclear whether there are either reductions or increases in NE production occurring in specific brain regions which could be a contributing factor.

### 4.3. Serotonergic Signaling in the Brain

The majority of serotonergic input to the forebrain is provided by the dorsal raphe nucleus (DRn), which projects to the VTA [141] (Figure 2D). Inhibitory projections from the VTA and striatum innervate the DRn to link 5-HT and DA signaling [142], such as the DRn neurons inducing a bursting pattern in the VTA to drive behavioral reinforcement [143]. The DRn also projects to the cerebral cortex, limbic system, striatum, thalamus, and hypothalamus to influence mood, cognition, attention, and emotional regulation. Therefore, decreased 5-HT levels in the ADHD brain likely underlie emotional symptoms due to perturbed signaling in these pathways. 5-HT is metabolized from tryptophan, and increased levels have been observed in the plasma, serum, and urine of ADHD patients [144]. This seems to suggest that tryptophan is not being effectively converted to 5-HT, and dysfunction of genes involved in the metabolic pathway has been associated with ADHD [145]. PET imaging has also been used to assess the levels of the serotonin transporter (SERT) in the brains of ADHD patients. Vanicek et al. did not find significant differences in SERT availability across brain regions between ADHD adults and TDIs [146]. These findings aligned with a previous study that also found no differences in SERT density linked to ADHD [147]. Therefore, it appears that 5-HT metabolism may be another contributing factor to ADHD pathophysiology; however the precise target sites and the underlying mechanisms remain poorly defined.

### 4.4. NMDAR and Dopamine Interactions

While ADHD medications primarily target monoamine signaling, it is important to discuss how this may contribute to other signaling pathways that are integrally involved in cognition. There have been well-characterized interactions between D1-like and D2-like receptors with N-methyl-D-aspartate receptors (NMDARs) in the cortex, hippocampus, and striatum [148] (Figure 3). NMDAR is a glutamate receptor that plays a key role in synaptic plasticity, which is thought to form the basis of learning and memory [149]. Activation of NMDAR is dependent on both glutamate binding, alongside depolarization from α-amino-3-hydroxy-5-methyl-4-isoxazole propionic receptors (AMPARs), which removes the magnesium ions that block the NMDAR channel pore [149]. This enables calcium influx, activating a molecular cascade which can increase cAMP levels, leading to activation of PKA. Therefore, given that several of the molecules involved in NDMAR signaling are shared with DA receptor activation, there are many implications for ADHD medications and synaptic plasticity.

Modulation of NMDAR by these medications appears to be dependent on the NMDAR subunit composition. For instance, GluN2A-NMDAR predominant entorhinal-CA1 synapses show depressed NMDAR excitatory postsynaptic potentials (EPSCs), whereas GluN2B-NMDAR predominant CA3-CA1 inputs have potentiated NMDAR EPSCs from D1-like receptor activation [150]. For D2-like receptors, modulation of NMDAR appears more straightforward; activation of D2 receptors tends to depress NMDAR activity. Altogether, this produces a complex system by which ADHD medications can both enhance or depress NMDAR-mediated plasticity, which will be covered when we discuss the impact of these medications on cognition.

## 5. Genetic Contributions to ADHD

ADHD has long been considered highly heritable, with analyses of twin and family studies estimating its heritability between 74–80% [151,152]. Children of parents with ADHD have an elevated risk of having full or subthreshold ADHD than children born to parents without ADHD [153]. Given the scope of ADHD genetics, here we highlight key findings and direct readers to the following reviews for additional context [151,154]. Observations of ADHD-typical behavior in syndromic disorders and multiple studies have found a strong genetic association with altered neurodevelopment and structural and functional brain changes in ADHD [151,152,155].

Many of the genes historically associated with ADHD pertain to monoaminergic signaling. This includes mutations in genes encoding monoaminergic transporters, such as the *SLC6A2* gene, which encodes for NET, and has been linked to deficits in visual memory and attention [134,156,157]. Similarly, mutations in the *SLC6A3* [158,159,160] and *SLC6A4* genes [161,162], which encode for DAT and SERT, respectively, have also been associated with traditional ADHD symptoms. This extends towards genes which synthesize enzymes involved in monoamine metabolism, such as *DBH* [138,139,140] and *MAOA* [163], alongside genes that synthesize the monoamine receptors themselves, including *DRD1* [164,165], *DRD2* [166], *DRD4* [167,168,169], *DRD5* [170,171], *ADRA2A* [172,173], and *HTR1B* [174,175,176,177]. This again points towards monoaminergic dysfunction being a contributor to ADHD development and symptoms. Given that monoamines primarily act as neuromodulators, it is therefore not surprising that many non-monoaminergic genes involved in broader processes such as neurodevelopment, synaptic plasticity, and neuronal migration are also implicated in ADHD [152]. For example, *CDH13* encodes a cadherin involved in neuronal signaling, neurite growth, and synapse development, and has been linked to hyperactivity and impulsivity symptoms characteristic of ADHD [178,179,180]. Other genes linked to ADHD include *CTNNA2* which encodes an alpha-catenin protein involved in neuronal migration and synaptic plasticity [181,182,183], and *SORCS3* which influences postsynaptic depression and hippocampal-dependent learning [184,185,186]. Given the interplay between neuromodulation and synaptic machinery, it remains challenging to deduce the relationship between monoaminergic dysfunction and synaptic plasticity at the genomic level for ADHD.

The precise genetic mechanisms underlying ADHD remain unclear [187], as no specific gene variant has been found that directly predicts ADHD [152]. Genetic linkage studies have demonstrated that the effect of SNPs only explains 22% of ADHD heritability, which is markedly lower than the estimated heritability of ADHD [151]. This discrepancy may reflect variable methodology, the influence of established environmental risk factors for ADHD, and likely that the nature of ADHD heritability is largely polygenic [151,187,188]. While polygenic risk scores can predict ADHD, there is concern that they may not necessarily be causative or generalizable, and that studies are underpowered and limited in the number of ethnicities genetically analyzed [153,189]. Altogether, these findings highlight that ADHD is a highly polygenic and multifactorial disorder, with genetic risk shaped by numerous small-effect variants in combination with environmental influences. A summary of these genes is available in Table 2.

## 6. ADHD Medications

As outlined above, modulation of monoaminergic pathways is likely to underlie the efficacy of ADHD medications. The guidelines for prescribing ADHD medications differ between countries. For instance, in the United Kingdom, MPH is typically offered as the first-line pharmacological treatment for children aged ≥5 or adolescents with ADHD under brand names like Ritalin or Concerta [203]. However, in North America, both AMPH and MPH-based medications can be considered as first-line treatments [204,205]. In general, assessments are performed to determine the cardiovascular health of the patient before stimulant medications are considered, in addition to discussing the preference for the possible side effect profile of the different medications. This includes taking a family history of heart disease, as stimulant medications increase blood pressure by enhancing NE and DA signaling. A general overview of the prescription of these medications is provided in Figure 4.

## 7. Modes of Action of Various ADHD Medications

While ADHD medications are somewhat similar in accomplishing the same end goal of elevating monoamine levels, their mechanisms are unique and underlie their differences in potency and side effects. These mechanisms for stimulants and non-stimulants are summarized in Figure 5.

### 7.1. Methylphenidate Based Medications

MPH-based medications function to inhibit DAT and NET, increasing the availability of DA and NE in the synaptic cleft (Figure 5A) [206]. This results in an increase in dopaminergic and adrenergic signaling within key brain regions such as the PFC, striatum, and NAc, to enhance memory, attention, motivation, motor control, and improve reward-seeking behavior, respectively [207,208]. However, MPH does not significantly impact the SERT, thus there is minimal change in the availability of 5-HT within the brain as a direct effect of the drug [209,210]. Additionally, it appears that MPH also alters the distribution of dopamine vesicles, both in shifting them closer to active zones in the presynaptic terminal, or back to the cytoplasm for later release [211,212]. While this mechanism is not fully understood, it is possible that an elevation of DA through DAT inhibition may cause the recruitment of DA vesicles dependent on dopaminergic activity. This potentially identifies a mechanism by which MPH enables more controlled release of DA as compared to AMPH.

### 7.2. Amphetamine Based Medications

AMPH-based medications differ from MPH and non-stimulants in that they enter the presynaptic neuron using DATs (Figure 5B). Unlike MPH, AMPH directly inhibits the vesicular monoamine transporter 2 (VMAT2), which impairs the packaging of monoamines into vesicles for subsequent release into the synaptic cleft. Concurrently, AMPH also impairs monoamine oxidase (MAO), preventing the breakdown of monoamines [213,214]. As a result of these two mechanisms, AMPH increases the concentration of monoamines in the presynaptic neuron, leading to a reversal of the monoamine transporters to shuttle DA, NE, and 5-HT into the synaptic cleft [215]. Like MPH, this results in increased alertness and concentration to reduce symptoms of ADHD. However, AMPH also increases 5-HT transmission by blocking the SERT [216], which could theoretically contribute to modulating DA release through RNc-VTA signaling [217] and in turn contribute to improving mood-related symptoms of ADHD.

### 7.3. Non-Stimulant Medications

Non-stimulants, on the other hand, do not directly increase DA levels. For example, ATX is a selective NE reuptake inhibitor, which blocks the NET and increases the levels of NE, increasing availability in the synaptic cleft (Figure 5C) [218]. As it does not directly increase DA levels, ATX often produces less severe side effects while reducing abuse risk linked to DA modulation of reward pathways [219]. Within the PFC, NET also partially contributes to reuptake of DA [220], thus ATX does still provide indirect benefits for increased DA transmission [221,222,223]. Viloxazine, a more recently approved ADHD medication, similarly inhibits the reuptake of NE which also increases DA levels in the PFC [224]. However, viloxazine differs from other non-stimulants in that it also increases 5-HT transmission through binding to 5HT2C receptors (Figure 5D). It has also been shown that viloxazine antagonizes 5HT2B receptors on GABA interneurons to increase the availability of 5-HT in the prefrontal cortex [225]. Guanfacine (Figure 5E) and clonidine (Figure 5F) are two other non-stimulant medications that function to modulate NE transmission. These medications bind to α2 adrenergic receptors, providing negative feedback thus reducing excessive NE release [226]. Guanfacine specifically targets α2A receptors located in the PFC and LC, which results in improved attention, working memory, and impulse control [227,228]. Clonidine, on the other hand, binds to α2B receptors in the peripheral blood vessels to regulate vasoconstriction, and α2C receptors in the basal ganglia and limbic system which regulates mood and startle response [226]. Thus, clonidine calms hyperarousal while improving sleep issues or anxiety and irritability.

## 8. Pharmacological Treatment for ADHD in Humans and Animal Models: Discrepancies and Possible Explanations

Animal studies form the backbone for drug discovery, for studying the effects of long-term exposure to various ADHD medications, and long-term toxicology at the preclinical stage. However, findings from animal studies do not always translate into clinical outcomes. This may be due to the heterogeneity of ADHD pathophysiology in humans, which is not fully represented in strain-controlled animal models that are devoid of several behavioral traits that are distinct to humans. The studies on “ADHD” animal models predominantly focus on modifying the genome to induce ADHD-like symptoms [229], whereby the genetic makeup and the behavioral repertoire may not match that of humans. While these studies may still be useful for testing the impact of specific molecular mechanisms in augmenting select “symptoms”, or testing the efficacy of ADHD medications, the etiology of human ADHD itself is not always from a homogenous genetic cause. For example, DAT mutant mice do demonstrate “ADHD-like” symptoms, and present excessive dopamine levels [230]. However, while excessive DA levels are present for some ADHD patients, in others, there is a decrease in DA signaling [74]. Similarly, the spontaneously hypertensive rat model has been used to model human ADHD. These animals demonstrate core ADHD symptoms in hyperactivity, impulsiveness, and inattention, and are more sensitive to delays in reinforcement like ADHD children [231]. While this model is effective for exploring pharmacological mechanisms in relation to behavior, hypertension itself is not a direct symptom of ADHD, and may produce confounding effects on behavior and cognition through influencing blood flow to the brain [232,233,234]. This is to say that it is unlikely that a single animal model of ADHD could recapitulate the behavioral repertoire that forms the hallmarks of ADHD in humans. Even if this were the case, then the broad heterogeneity seen in human ADHD individuals would make the phenotype indiscernible. An overview of commonly used ADHD animal models is provided in Table 3.

For prenatal exposure specifically, it is also important to consider the significant differences between rodent and human development. Rodents are typically considered to have reached adulthood around ~70 days following birth [250]. Humans on the other hand continue brain development until around 25 years of age [251]. As a result, we argue that this expedited neurodevelopment in rodents relative to humans may increase the susceptibility to perturbations from prenatal ADHD medication exposure. While humans experience greater overall exposure to these medications throughout their lifetime, rodent models may instead involve a greater proportion of their developmental timespan. Moreover, the liver function and potential metabolic breakdown of various medications may also vary between rodents and humans, altering the pharmacokinetics and pharmacodynamics of these drugs [252]. This might help explain why rodent models of prenatal exposure to ADHD medications demonstrate adverse effects not seen in humans. However, we also argue that this would make it even more challenging to ascertain the long-term effects of exposure to these medications in humans. Assessing the long-term effects of ADHD medications in humans would likely require longitudinal studies tracking cognitive function and MRI imaging from childhood to adulthood and older age, which raises feasibility concerns. There would also be a need to compare the relative change in cognition from timepoints between ADHD patients and TDIs, since it has been shown that ADHD medications do not always fully restore cognitive function [253]. Given that this is likely unrealistic from a cost and logistical perspective, animal models still provide us with valuable insights into the long-term effects of these medications. For example, there continues to be novel mechanisms being deduced that underlie the efficacy of amphetamine for treating ADHD, despite the medication being already prescribed for decades [254]. Even newer ADHD medications like Viloxazine, had their fundamental mechanisms deduced using animal models [255,256]. Animal models enable interrogation of molecular mechanisms to be made that would not otherwise be possible in humans, especially when considering in utero exposure. Here we will discuss the implications of treatment of human ADHD with AMPH- and MPH-based medications, alongside ATX for cognition, and discuss how these findings relate to what has been shown in animal models of ADHD.

### 8.1. ADHD Medications Generally Improve Cognition in Humans

Much of the controversy regarding ADHD medications pertains to concerns regarding their abuse as cognitive enhancers, alongside the potential for long-term side effects when prescribed to children. In general, it appears that ADHD medications do not always produce an immediate improvement in cognitive performance for individuals with ADHD, where long-term use of the medication over several weeks is required to see improvements in ADHD symptoms [257]. While these medications often improve memory for individuals with ADHD, they do not do so to the level of TDIs [258]. A systematic review which included 176 studies from 1980 to 2012 concluded that long-term academic outcomes for children with ADHD are improved with ADHD medication, but multimodal treatment provided the best results [259]. However, a study by Morell & Expόsito concluded long-term treatment with either stimulants or non-stimulants does not necessarily always improve all ADHD symptoms, such as delay aversion and risk-taking behavior [253]. This suggests that it may still be worthwhile for medical practitioners to explore treatment plans that also include non-pharmacological approaches, to further reduce the gap in learning outcomes between children with ADHD and their classmates. Additionally, it is crucial to consider how the side effects of ADHD medications may indirectly perturb learning, memory, attention, ability to multitask, and general cognition. For example, impaired sleep quality resulting from these medications [260] or stressful settings, might impair their encoding of memory [261]. Overall, ADHD medications, when taken as prescribed, do appear to improve performance in academic and work settings for individuals with ADHD. Additionally, stimulant-related ADHD medications have also been found to decrease the aforementioned alterations in brain structure associated with ADHD [262,263]. However, these effects appear to be age-specific, for example, MPH has been shown to reduce cortical thinning in children with ADHD but not in adults [264]. This suggests that treatment with these medications in childhood may provide long-term benefits for brain maturation and trajectory, and thus cognition. On the other hand, cumulative exposure to ADHD medications has been linked to decreased hippocampal CA1 region volumes in children [265], but it remains unclear why these side effects appear to be region specific and how this reduced hippocampal volume might correspond with learning and memory in ADHD children. Notwithstanding the potential beneficial effects of various medications, there exists a possibility of off-target effects of these compounds, and these are perhaps the least studied.

### 8.2. Amphetamine Effects on Cognition

For AMPH specifically, the effects of the drug on different types of cognition have had conflicting results, particularly when comparing human and animal studies. For individuals with ADHD, AMPH is often reported to enhance aspects of cognition, but these improvements appear to be highly individualized and context dependent. A systematic review from 2022 [266] highlighted that only one [267] out of the nine studies they looked at regarding the effect of AMPH on visual working memory (VWM) found improvements from the drug. This is likely due to differences between the studies in the time between participants taking AMPH and performing the VWM test. The study by de Wit et al. [267] had participants perform the VWM test around 3 h after taking AMPH, which would be the time period in which AMPH generally reaches the highest peak drug concentration in the bloodstream [268]. Additionally, participants within the de Wit et al. study conducted a baseline assessment on VWM on the same day on which they completed the treatment test for VWM; both factors might explain why their study found improvements in VWM, whereas the other studies failed to see any significant improvements. These inconsistencies in the effect of AMPH were also true for spatial working memory (SWM). Within this systematic review [266], four out of 13 studies reported AMPH-based improvements in SWM [269,270,271,272], and three of these highlighted time-based effects on AMPH improvement of SWM [273,274,275]. These inconsistencies may be partly explained by factors such as that individuals with ADHD tend to have worse test-taking performance [276] while exhibiting increased test-anxiety [277]. Given that DA has been shown to have an inverse U-shaped action on human working memory and cognitive performance [278], it is possible that AMPH, through rapidly increasing extracellular DA, may instead impair memory performance under stress. AMPH is shown to improve academic performance for children with ADHD [279], but not necessarily in all areas of cognition. Hence, it is likely that AMPH improvements in cognition are highly individualistic, and contingent upon specific brain chemistry of the individual in question. In general, it seems that these medications do not inherently improve academic performance. For example, non-prescribed use of ADHD medications has been linked to impaired academic habits, such as studying and motivation during cognitive testing [280]. Therefore, while AMPH does often enhance cognition, this will not always manifest itself in academic outcomes or assessments of cognitive abilities. It also remains unclear whether such improvements tap into short-term working memory or if they also culminate into long-term cognitive enhancement.

In contrast, animal studies offer mechanistic insights regarding how AMPH influences cognition at the cellular and synaptic level, particularly in the hippocampus, a key region for learning and memory processes. Acute AMPH exposure has been shown to enhance synaptic plasticity through an increased phosphorylation of CREB [281] and AMPAR GluA1 subunits in the CA1 and dentate gyrus regions [282], suggesting short-term enhancements in memory formation. However, chronic exposure to AMPH and sensitization to the drug appear to impair synaptic plasticity in animal models. Repeated AMPH administration has been linked to impairments in both short- and long-term memory in the hippocampus, reduced neuronal density in the CA1 region, and morphological changes such as longer dendrites with fewer mature spines [283]. Given that mature spines are thought to be associated with increased long-term potentiation (LTP) and memory [284], the loss of these mature spines may be the mechanism by which AMPH sensitization impairs hippocampal-based memory. Interestingly, AMPH can also promote neurite outgrowth in vitro through pathways independent of DA or nerve growth factor release [285], and yet chronic AMPH exposure appears to reduce both nerve growth factor and brain-derived neurotrophic factor (BDNF) levels in key brain regions, suggesting a potential downregulation of neurotrophic support over time [286].

AMPH has been shown to increase NMDAR-GluN2B synaptic currents in midbrain dopaminergic neurons [287]. This potentiation does not, however, occur when DAT is inhibited; thus, the enhancement of NMDAR-mediated plasticity appears to be downstream of the established mechanism of AMPH entering the presynaptic terminal. Animal models have revealed that withdrawal from repeated AMPH administration can cause a downregulation in the expression of NMDAR [288], GluR1, GluR2 [289], and the GluN2B subunit [290]. This seems to suggest that withdrawal from AMPH-containing ADHD medications may cause a rebound effect in decreasing excitatory transmission. Moreover, the reduction in NMDAR expression may contribute to impairments in cognition that the ADHD individuals experience when stopping their medication [291].

Chronic AMPH exposure itself has been shown to cause downregulation of GluN2B, leading to long-term depression at cortico-accumbal glutamatergic synapses, resulting in behavioral sensitization to AMPH [290]. Interestingly, it appears that a loss of NMDAR signaling with specifically D1-like receptors may prevent AMPH sensitization [292]. When NMDAR signaling was lost with both D1R and D2R expressing neurons, sensitization to AMPH occurred [292]. This suggests that NMDAR contributes to synaptic plasticity with D2R-expressing neurons, to ensure that they could provide sufficient inhibitory control over excessive DA to prevent addictive behaviors associated with AMPH.

Altogether, these findings from both human and animal studies converge on a central theme: the effects of AMPH on cognition are highly context dependent. Acute exposure may provide benefits for cognition through enhancing synaptic plasticity, particularly under optimal arousal conditions. However, chronic use and sensitization may instead induce neuroadaptive changes that impair memory and neural resistance. This underscores the importance of considering both neurobiological and contextual variables when evaluating the efficacy of AMPH for enhancing cognition, particularly in clinical settings.

### 8.3. Methylphenidate Effects on Cognition

MPH has been shown to exert inconsistent effects on different forms of cognition. While MPH improves spatial memory during an initial assessment, it paradoxically leads to decreased performance accuracy upon subsequent examination [293]. These findings are in line with other studies that have demonstrated MPH-based improvements in SWM, sustained attention, and working memory tasks [294,295]. However, other studies have found that MPH does not improve spatial working memory nor the integration of sensory information [296,297]. Notably, despite MPH-induced increases in sympathetic arousal, emotional memory consolidation appears to remain unaltered [298].

Therefore, the question remains as to why, even with MPH treatment, individuals with ADHD still, on average, do not reach the same level of academic achievement as compared to TDIs. One possibility is that MPH does not fully normalize intrinsic motivation or account for the comorbidities of learning disabilities, anxiety, depression, or sleeping disorders associated with ADHD. Moreover, the side effects of MPH which include reduced appetite and insomnia, may in some cases attenuate the functional benefits conferred by increased DA and NE signaling, thereby limiting improvements for some individuals with ADHD in academic and work settings.

At the molecular level, repeated MPH exposure produces sustained repression of CREB in the NAc when examined in an animal model [299]. This is likely linked to other findings that repeated MPH treatment represses PKA activity, alongside DA-induced adenyl cyclase activation in the DS [300]. These findings are further supported by evidence of downregulation of plasticity-related gene expression in the striatum following chronic MPH treatment [301]. Additionally, repeated MPH exposure has been shown to repress the expression of CREB, BDNF, PKA, MAPK3, and cFos in the amygdala [302], all of which contribute to molecular mechanisms governing synaptic plasticity. Together, these animal studies suggest that repeated MPH exposure may compromise plasticity in the amygdala and striatum, potentially posing risks for perturbed fear and reward responsiveness.

The above-observed changes in plasticity from MPH exposure in animal models appear to be linked to NMDAR. MPH has also been demonstrated to alter NMDAR responses [303,304], and rescue impaired synaptic plasticity in an NMDAR-dependent manner [305]. This notion alludes towards an alteration of NMDAR expression similar to AMPH, where a single dose of MPH has been shown to significantly reduce protein levels of NMDAR subunits GluN1 and GluN2B [306]. Similarly, chronic MPH administration has been linked to widespread reductions in NMDAR expression across brain regions [307]. Given that the D1-like activation of GluN2B and predominant NMDAR contribute to synaptic plasticity, it would be important to investigate whether MPH may inhibit NMDAR plasticity for neurons expressing D1-like receptors. By downregulating these receptors, MPH may indirectly limit the capacity for plasticity in dopaminergic circuits, potentially contributing to reductions in impulsivity at a cost to cognitive flexibility or learning capacity.

The aforementioned molecular alterations are reflected in behavioral outcomes: chronic MPH exposure impairs object recognition in adolescent rats, but not in adults [308]. This same study reported the presence of depressive symptoms in adult rats, but not in adolescent rodents exposed to MPH. One explanation for this might be that early treatment of ADHD with MPH in human children is thought to normalize brain volumes [262]. Thus, it may be the case that repeated MPH exposure in young brain networks improves mesolimbic signaling in a manner that reduces depression symptoms, but the prolonged elevation of these transmitters would likely impair memory processes.

### 8.4. Atomoxetine Effects on Cognition

Like AMPH and MPH, ATX has generally been demonstrated to improve cognition for individuals with ADHD. This includes improvements in pattern and spatial recognition memory [309], reduced anxiety symptoms [310,311], and improved response inhibition [312]. Interestingly, ATX is known to reduce cognitive difficulties in perimenopausal and postmenopausal women with ADHD-like symptoms [313], suggesting that its utility may extend beyond traditional pediatric or young adult populations. However, several studies have shown that ATX does not always improve all cognitive challenges associated with ADHD, such as sustained attention [312] and visuospatial working memory [314]. Additionally, ATX has been shown to differ from MPH in how it modulates reward-based working memory. It appears that relative to MPH, ATX decreases the activation of working memory networks to reduce the typical enhancement of working memory when a reward is expected [315]. This might suggest that for individuals with ADHD who rely heavily on external motivation like rewards, ATX may not be as effective as MPH. On the other hand, it is also possible that ATX might be more effective than MPH in low-stimulation or low-reward environments by regulating the brain without conflicting with motivational systems.

Animal models have thus helped elucidate the neurobiological mechanisms underlying these cognitive effects of ATX. For instance, repeated exposure to ATX has been demonstrated to increase mRNA levels in the hippocampus and PFC, while enhancing phosphorylation of AKT and GSK3B which are downstream components of BDNF signaling [316]. At the protein level, there was greater BDNF present in the PFC, which suggests ATX facilitates cognitive enhancement through enhancing synaptic plasticity via the CREB pathway.

Beyond BDNF-mediated plasticity, ATX also appears to enhance cholinergic neurotransmission within the PFC and hippocampus through α1 and D1 receptor activation [317]. These neurochemical changes are reflected behaviorally: whereby animals treated with ATX exhibit improved performance in memory tests [317], consistent with other studies demonstrating that ATX facilitates learning and memory in preclinical models [318,319,320]. Interestingly, ATX has been shown to increase histamine release, which improved performance in the Morris water maze test [320]. Given the role of histamine as a modulator of learning and memory [321], this might suggest another mechanism by which ATX improves symptoms of ADHD and enhances cognition.

There are fewer studies on ATX and NMDAR. ATX is shown to rescue LTP through a postsynaptic mechanism [318]. Similar findings to MPH and AMPH have been observed with decreased expression of GluN2B, alongside reduced mRNA levels for genes encoding for NMDAR [318,322]. ATX, MPH, and D-AMPH all increase firing of PFC neurons through potentiating NMDAR activation [323].

Together, these findings underscore the multifaceted mechanisms of ATX across various cognitive domains. Mechanistically, ATX appears to exert its effects through a variety of pathways involving NE, BDNF signaling, cholinergic modulation, and possibly histaminergic activity. Continued integration of human and animal studies will be essential to fully elucidate the therapeutic scope and neurobiological impact of ATX.

Given that chronic exposure to ADHD medications reduces NMDAR expression, it is crucial for future studies to investigate the extent to which this impacts long-term learning and memory outcomes. This decreased NMDAR plasticity might be an adverse side effect of these medications, or it could simply be a mechanism by which they exert their therapeutic effects. It is possible that this reduced NMDAR plasticity, if targeted towards reward circuits, could be protective against impulsive behavior.

Altogether, animal models have revealed that prolonged exposure to ADHD medications may impair cognition. However, it remains unclear whether these findings are translatable to humans since there are significant differences in human metabolism of these medications as compared to the rodent models. It is understandably challenging to perform long-term studies exploring the implications of prolonged ADHD medication use from adolescence to adulthood. There are many confounding variables that would need to be accounted for regarding environmental and social factors. As of right now, it appears that long-term exposure to ADHD medications in humans is potentially safe and that these treatments do provide benefits to individuals. It is crucial that we continue to explore the implications of prolonged exposure through animal studies and pursue longitudinal studies in humans to evaluate how chronic use of these medications alters brain connectivity, activity, and behavior.

## 9. Implications of In Utero Exposure to ADHD Medications for Long-Term Cognition

### 9.1. Prescription of ADHD Medications During Pregnancy

One of the outstanding questions is whether ADHD medications are safe to use during pregnancy. Both AMPH and MPH have been shown to cross the placental barrier in humans and in rodents [324,325,326,327], and ATX in rats specifically [328]. Under most circumstances, pregnant individuals with ADHD are discouraged from taking these medications during pregnancy [329]. However, stopping psychostimulant treatment for ADHD during pregnancy has been shown to increase the risk of depressive symptoms in patients [330], in addition to the potential cognitive and functional implications on productivity. Therefore, psychostimulants may still be prescribed during pregnancy, depending on the risk-to-benefit ratio, and these clinical considerations have been covered in a recent review [331]. ADHD medication use during pregnancy increased from 0.3% in 1998 to 1.3% in 2014 in North America [332]. Additionally, the percentage of privately insured reproductive-aged individuals who filled ADHD medication prescriptions increased from 0.9% in 2003 to 4.0% in 2015 in the United States [333]. Thus, with the increasing use of these medications during pregnancy and for woman of childbearing age, there is a dire need to validate their safety for developing the fetal brain; and this would require a paradigm shift in our approach to ADHD medication.

### 9.2. Continued Evidence Suggests That ADHD Medications Are Safe During Pregnancy

Regarding concerns of birth complications, developmental defects, and risks of neurodevelopmental disorders, human studies have largely provided reassurance that ADHD medication use during pregnancy is safe [334,335]. Most recently, a population-based cohort study followed 2257 children exposed to ADHD medications during pregnancy, for a mean follow-up time of 7 years [336]. Their findings highlight that AMPH, MPH, and ATX did not increase risks for long-term neurodevelopmental disorders. Some studies have however found a small increase in cardiac malformations associated with prenatal exposure to MPH, alongside decreased birth weights with both MPH and AMPH [337,338]. Moreover, prenatal exposure to AMPH has been linked to impaired academic outcomes [339], alongside increased symptoms of depression, anxiety, and ADHD [327]. However, much remains unknown regarding how in utero exposure to these medications may alter neurodevelopment from a molecular perspective. Animal models are often used to explore the impact of drug exposure on neurodevelopment, where postnatal day 0–14 in rodents is considered comparable to the third trimester in humans [340]. Here, we will review the cellular, molecular, and behavioral changes caused by prenatal exposure to ADHD psychostimulants in animal models, as few studies have been performed using non-stimulants.

### 9.3. In Utero Exposure to AMPH: Time-Dependent Changes in Metabolism and Behavior in Animal Studies

Prenatal AMPH exposure is linked to decreased expression of D2-like inhibitory receptors in the NAc, alongside reduced expression of tyrosine hydroxylase [341]. Loss of these D2-like inhibitory receptors increases the risk of addictive behaviors in adulthood [342,343]. Thus, there is the possibility that prenatal exposure to AMPH may impair inhibitory signaling within reward pathways in the brain, posing risks for addiction behaviors later in life. Additionally, several studies have shown that prenatal AMPH exposure alters the metabolism and levels of monoamines in the brain [344,345,346]. One study, which looked at whole-brain concentrations, reported increased 5-HT levels at birth that were normalized to those of controls over subsequent days [344]. This also included higher levels of GABA on postnatal day 21, which correlated with decreased locomotor activity in these prenatally exposed pups. This reduction in motor activity from prenatal AMPH exposure has also been reported in 45-day-old offspring [347]. Therefore, in animal models at least, prenatal AMPH exposure may have long-lasting effects on activity, likely resulting from increased inhibitory transmission.

On the other hand, reduced 5-HT levels have been shown in the medial PFC with prenatal AMPH exposure [346]. These offspring also exhibited reduced neuronal density within the medial PFC shortly after birth, which normalized by day 30. This suggests that some perturbations caused by prenatal AMPH exposure may not be permanent, and recovery may ensue in the weeks following birth. For example, offspring of AMPH-exposed rats were shown to exhibit more memory errors in the Lashley III maze test 90 days following birth, but these errors normalized to those of controls in subsequent assessments [348]. The alterations in monoamine metabolism appear to extend towards increased conversion of tyrosine into DA and NE [345], which were linked with improved cognitive processes at 90 days post-birth [349]. It has also been shown that prenatal AMPH exposure in *C. elegans* causes increased expression of tyrosine hydroxylase paired with decreased expression of VMAT in the adult animal [350]. These same animals also demonstrated increased sensitivity to AMPH. One possibility could be the loss of VMAT, which may have further potentiated AMPH-mediated reverse transport of monoamines. These findings have important potential implications for ADHD medication use during pregnancy. Given that ADHD can often be hereditary, if the pregnant individual is prescribed AMPH for their ADHD, it is possible that their child may be prescribed a similar medication in the future. Thus, these findings suggest that AMPH use during pregnancy may increase the offspring’s sensitivity to the drug, thus making them more susceptible to ADHD symptoms. Overall, it appears that the alterations in monoamine metabolism caused by prenatal AMPH exposure in animals are likely temporary, and do not produce life-long impairments in behavior.

However, prenatal exposure to AMPH is known to cause life-long impairments in glucose metabolism. Prenatal AMPH exposure was linked to a reduction in 5-HT within β-cells of the pancreas, alongside decreased expression of markers of pancreas maturation [351]. Mice of both sexes prenatally exposed to AMPH exhibited hypoglycemia 1 year after birth. However, female mice demonstrated sex differences with reduced β-cell islet sizes, alongside decreased bodyweight as compared to controls. This may stem from previous evidence pointing towards estrogen increasing the sensitivity of female rats to AMPH [352]. Therefore, the presence of sex differences is interesting given that during prenatal exposure to the drug, both male and female animals are in a similar environment. It is plausible that sex-differentiation of the brain via estrogen in females may modulate the effects of AMPH in a manner that does not occur in males. Therefore, it is worth exploring whether prenatal exposure to lower concentrations of AMPH in female animals produces similar effects to standard concentrations in male animals.

### 9.4. In Utero Exposure to Methylphenidate: Alterations in DA Pathways and Behavior in Animal Studies

Like AMPH, prenatal MPH exposure alters the expression of D2-like receptors. Adult male rats prenatally exposed to MPH present increased expression of tyrosine hydroxylase in the SNc, alongside NAc core and shell regions [353]. Both the dorsal and ventral regions of the striatum presented an increase in DAT density, whereas the SNc and VTA showed an increase in D2R density. This suggests that prenatal exposure to MPH increases DA metabolism paired with elevated regulatory control of DA reuptake and signaling, which manifests in reduced susceptibility to hyperactivity caused by cocaine. Similar findings have been reported with prenatal MPH exposure increasing time to seek rewards [354], suggesting reduced impulsivity. However, this was a sex-specific effect for males, whereas females demonstrated decreased time to seek rewards. It is unclear what causes this sex-specific effect, but one possibility is that prenatal MPH exposure may alter the development of the PFC in females to facilitate more impulsive decision-making. Again, it may be the case that even though the same concentration of the drug was used across both male and female animals, estrogen may have potentiated the effects of MPH in females to cause these unintended symptoms. The evidence that increased impulsivity is observed in females is supported by the findings that fetal MPH exposure induces increased locomotor activity and decreases anxiety behavior [355]. Therefore, at least in animal models, it appears that prenatal MPH exposure may induce ADHD-like behavior in the offspring and produce more pronounced effects in females as compared to males.

The impact of prenatal MPH exposure on behavior also appears to be dependent on when the drug is taken during pregnancy. MPH exposure on embryonic days 8–10 results in decreased anxiety-related behaviors and an increase in exploratory behavior that did not occur after exposures on embryonic days 12–14, 16–18 [356]. This suggests that exposure to MPH during early fetal development can have more significant effects on behavior. When considering exposure even earlier in development, such as 0–5 days post-fertilization, it has been shown that MPH exposure can cause long-term impairments in spatial learning and predatory avoidance responses [357]. Altogether, evidence from animal models suggests that prenatal MPH exposure may cause ADHD symptoms with temporal and sex-specific influences. Given that prenatal MPH exposure in humans does not appear to increase the likelihood of developing ADHD [336], more research is required to determine whether these findings from animal models are applicable to the human fetal brain. It is possible that the observed effects in humans are either more subtle or take a longer time to manifest (such as adulthood) or potentially rectified through neuroplasticity, and as such remain undetected in clinical studies.

Considering the side effects of many pharmacological agents on the brain and behavior of ADHD patients, alternative and non-pharmacological approaches have come to light.

## 10. Transcranial Stimulation for ADHD Treatment

Given that ADHD pharmacotherapies are not always effective for all ADHD patients, non-pharmacological treatments are being explored. Transcranial stimulation has gained traction as a potential future treatment for ADHD. Modulation of monoamine pathways is believed to underlie the efficacy of repetitive transcranial magnetic stimulation (rTMS) and transcranial direct current stimulation (tDCS) (Figure 6). tDCS is a non-invasive procedure that utilizes weak electrical currents (typically 1–2 mA) to modulate membrane excitability [358]. The cathode is used to hyperpolarize the membrane potential, whereas the anode is used to increase depolarization. On the other hand, rTMS involves the generation of an electric current in the brain through repeated magnetic pulses, which can directly trigger neuronal action potentials [359]. This can lead to alterations in synaptic plasticity, where generally low-frequency pulses (1 Hz) reduce cortical excitability, whereas high-frequency stimulation (5–20 Hz) produces the opposite effect [360]. Both tDCS and rTMS can thus influence neurotransmitter release, alongside the plasticity of networks that are thought to be perturbed in ADHD.

In the case of ADHD, these techniques are largely used to target the PFC [361], where reduced activity in ADHD patients has been documented [362,363]. In a cohort of 64 children with ADHD, Cao et al. [364] demonstrated that rTMS, when combined with ATX treatment, showed greater improvements in ADHD symptoms and executive function than ATX treatment alone. Subsequent studies have found similar improvements from rTMS for children with ADHD, which provided further evidence regarding the potential of this treatment for pediatric ADHD [365,366]. These improvements in cognition are thought to be linked to better blood flow, enhanced neuronal activity, and potentially improvements in sleep patterns for children with ADHD [367,368]. However, the precise mechanisms underlying the efficacy of rTMS remain unknown, and further studies are required to validate the efficacy of rTMS as a treatment option for ADHD.

tDCS is often used to target the left dorsolateral prefrontal cortex (dlPFC), which has been shown to improve response inhibition, attention, working memory, and cognitive flexibility in ADHD patients [369,370]. The outcome was not however consistent when tDCS was used to target the right dlPFC [371]. However, it does appear that tDCS targeting the dlPFC improves PFC activity during working memory tasks for adolescents with ADHD [372]. The left dlPFC is targeted based on its involvement with executive functioning, a key brain area where ADHD patients show deficits [371]. While still not fully understood, it is thought that the positive effects of stimulating the dlPFC occur due to the dlPFC being a crucial site for dopaminergic effects of cognition [369]. Improving dopaminergic signaling improves inhibitory control and decreases impulsive behaviors. tDCS stimulation of mouse motor cortex slices may enhance long-term potentiation in an NMDAR-dependent manner, while increasing BDNF release within this brain region [373]. Like TMS, tDCS is more effective with repetition, as cognitive improvements from a single treatment last less than three days. Regarding safety, neither TMS nor tDCS have serious adverse effects reported to date, although common minor adverse effects were present, including itchy scalp, headache, and fatigue [371]. A word of caution here is that the human brain is a highly integrated system with its innate communication highways laid during development; the perturbation of one area using either TMS or tDSC may have lasting, and yet undiscerned effects on other regions of the brain.

As an alternative, the peripheral vagus nerve can be targeted by tDCS by surgically implanting an electrode with programmed frequency, duration, and charge [374]. VNS activates afferent fibers of the auricular branch of the vagus nerve, which leads to stimulation of the LC to increase NE release [375]. In animal models, vagus nerve stimulation (VNS) has also been shown to increase synaptic plasticity through increasing LC firing rates, modifying hippocampal activity, and altering gene expression, leading to increased release of DA, NE, 5-HT, acetylcholine, and BDNF [376]. Common non-surgical adverse effects of VNS include hoarseness, cough, and headaches, though these tend to improve with time [377]. VNS is currently used to treat drug-resistant epilepsy but might show potential in the future for ADHD treatment [374]. Because surgical implantation is invasive and poses serious risks, an alternative option is external trigeminal nerve stimulation (eTNS) which is carried out by placing electrodes on the forehead above the nerve. The trigeminal nerve projects to the LC; thereby eTNS appears to increase NE in the PFC and hippocampus. eTNS appears to act through stimulation of the brain stem, targeting the LC and raphe nucleus to increase NE and 5-HT release, respectively [378,379]. McGough et al. [380] demonstrated that a 4-week-long nightly treatment with eTNS improved ADHD symptoms in children with ADHD, which led to the FDA approval of the eTNS for pediatric ADHD [381]. These findings aligned with their previous study, which involved 24 ADHD children receiving eTNS for 8 weeks, and found improvements in measures of executive functioning [382]. Both studies showed improvements in impulsivity, hyperactivity, and attention that were statistically significant compared to a sham group. Common adverse effects were mild, consisting of headaches, fatigue, and an increased appetite [380,382]. While transcranial stimulation appears promising as a future avenue of treatment for ADHD, pharmacological therapies remain at the forefront of ADHD research and still require further validation in terms of their safety and efficacy.

## 11. A Paradigm Shift in ADHD Research: Integrating Synaptic and Systems-Level Neuroscience

### 11.1. Reframing ADHD Treatment: From Disorder to Divergence

The classification of ADHD itself as a disorder reflects a framework that views deviation from normative behavior as inherently pathological. While this clinical model has been essential for securing support, services, and research, it can also reinforce stigma and negative self-perception in children diagnosed with ADHD. By defining ADHD primarily through its divergence from majority norms such as attention span, impulse control, and activity levels, there is the risk of portraying these traits as inherently defective rather than contextually challenging. This is intertwined with the controversy regarding concerns about the overprescription of ADHD medications. If ADHD is only viewed as a ‘disorder’, we then naturally gravitate heavily towards pharmacology without addressing the social context. Medication can help children regulate their attention and behavior, and there is evidence to support the notion that these treatments do help ameliorate brain structural changes in ADHD. However, these treatments should be complemented by furthering the integration of different learning styles in classroom settings, such as disseminating concepts through different forms of media and interactive materials. As Frolli et al. [383] demonstrated using the Universal Design for Learning framework [384], a more individual-centered approach to learning can enhance reading, writing, and mathematical skills for children with ADHD. If paired with educators who are equipped with greater awareness of how to recognize positive qualities of their students with ADHD, this could improve learning experiences in classroom settings. In the long term, this may reduce the reliance on chronic use of pharmacological interventions for ADHD.

### 11.2. Improving ADHD Diagnosis and Interpretation of Human Data

Future ADHD diagnostic criteria should be refined further to better reflect the diverse presentations of the condition across sex and age, to enable personalized treatment plans only feasible when diagnoses are specific for ADHD subtypes. The discrepancy in diagnostic rates between males and females, as well as the difficulty in diagnosing ADHD in adulthood, suggests that an overemphasis on observable physical hyperactivity may limit diagnostic generalizability. Greater awareness for healthcare practitioners on internal hyperactivity symptoms such as restlessness, sensory overload, and mental overstimulation, especially in low-stimulation environments, may serve to improve the accurate diagnosis of ADHD. Furthermore, distinguishing ADHD from other neurodevelopmental disorders like ASD requires greater attention to patients’ perceptions of social interaction and their ability to engage in reciprocal conversations, which are subtle yet revealing features [36] that standard assessments often overlook. Mechanistic research in humans, such as PET imaging studies of DAT availability, is frequently misinterpreted to infer dopaminergic tone. This same caution must apply to investigations of NET and SERT, as transporter availability does not directly equate to neurotransmitter levels. Monoamines act through diverse GPCR subtypes with distinct behavioral and cellular effects, and their availability alone cannot be assumed to reflect functional neurotransmission. Given that ADHD pharmacotherapy can normalize brain structure [262,264] and alter transporter availability [120], future studies must also account for treatment history, as grouping treated and untreated patients together risks obscuring clinically significant differences.

### 11.3. Further Longitudinal Studies Are Required to Assess the Impact of In Utero Exposure to ADHD Medications on Cognition

When it comes to deducing the impact of in utero exposure to ADHD medications in humans, we believe continued longitudinal assessments will produce key findings. For example, Madsen et al. [336] conducted a longitudinal study on in utero exposure to ADHD medication, with children being followed up from age 3 to a mean follow-up time of 6.9 years. Since they found no increased risk of neurodevelopmental disorders for these children who would be around 10 years old, it would be interesting to see if this is still the case once these children reach adulthood. Additionally, it would be worthwhile, where possible, to assess other parameters of cognition, such as academic outcomes, alongside social behaviors, particularly during adolescence.

### 11.4. Considerations for Animal Models and In Utero Exposure Studies

The use of ADHD animal models should be interpreted with subtype-specific context in mind. For example, the DAT knockout mice model ADHD subtypes with excess extracellular dopamine [231], and because amphetamine relies on DAT to enter presynaptic neurons, its effects cannot be directly compared to controls. Regarding in utero exposure to ADHD medications, animal studies often lack clinical relevance due to inconsistent dosing and use of non-oral administration routes. As McDonnell-Dowling et al. [385] review, doses used in rodent studies frequently exceed human-equivalent exposures, and the Reagan-Shaw et al. [386] allometric scaling method could be used to improve translational accuracy. Additionally, the route of drug administration is crucial, where oral gavage, which more closely mimics human oral intake, is a more appropriate method for drug delivery in these models. Additionally, the impact of sex differences on drug effects must be better characterized. Estrogen, which enhances sensitivity to several ADHD medications [352], may modulate pharmacological effects during pregnancy. Ultimately, combining well-designed animal models with longitudinal human studies will provide a clearer picture of the developmental consequences of ADHD pharmacotherapy during pregnancy.

## 12. Conclusions

While there remains limited clinical evidence that long-term use of prescribed ADHD medications produces adverse effects, animal models have raised the possibility that molecular mechanisms involved in cognition may be perturbed by chronic use. These medications are generally effective in improving cognitive deficits in patients with ADHD and thus remain the first-line treatment. Studies in humans continue to provide evidence that prescribed use of these medications during pregnancy does not produce noticeable cognitive impairments in children, but further studies are required to assess long-term cognitive outcomes for children exposed to these medications in utero. We believe that it is imperative that further animal research is dedicated to ascertaining potential molecular perturbations involved in cognition that result from in utero exposure. Within these future studies, it is crucial that appropriate methods of drug administration are leveraged, clinically relevant doses are used throughout, and the role of estrogen in enhancing the potency of AMPH during pregnancy is considered. Human studies should continue to be conducted to assess the consequences of in utero exposure, with an emphasis on considering the long-term cognitive outcomes of the child in adolescence and adulthood. The data obtained from humans is promising in terms of safety both during pregnancy and for general long-term prescribed use of these medications; however, the limited studies assessing long-term cognitive outcomes make it unclear whether there remain possible consequences for cognition.

## Figures and Tables

**Figure 1 cells-14-01367-f001:**
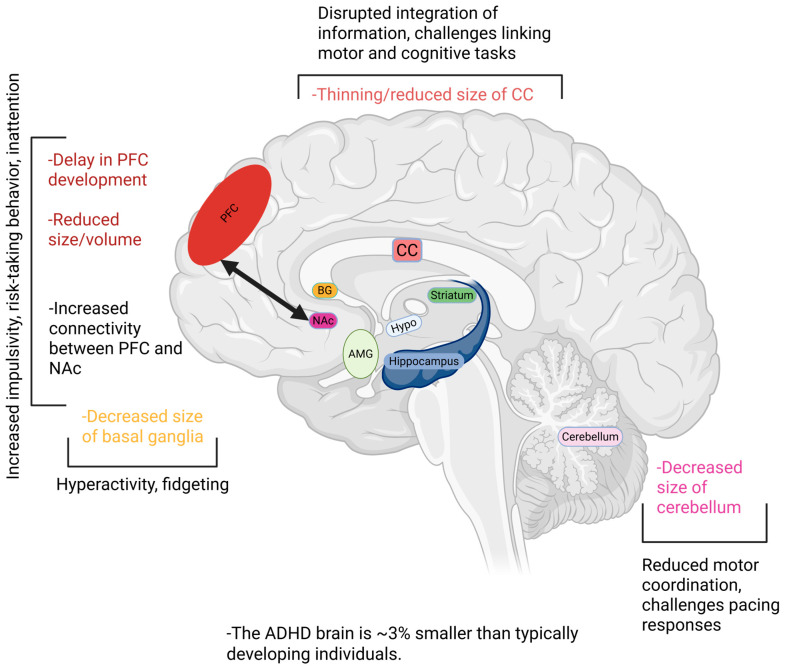
Brain regions that are altered in ADHD. Symptoms pertaining to impulsiveness and inattention may involve delayed PFC size during development, alongside increased signaling from the NAc. Impairments to information integration during tasks are linked to a reduced size and thinning of the CC, whereas reductions in motor coordination are associated with decreased volume in the cerebellum. The motor restlessness and fidgeting associated with ADHD as indications of hyperactivity are thought to arise from reduced size of the BG, which is crucial for regulating voluntary motor movements. In general, patients with ADHD are thought to have an approximately 3% decrease in overall brain volume. Created in BioRender. Yacoub, M. (2025) https://BioRender.com/j31x82e.

**Figure 2 cells-14-01367-f002:**
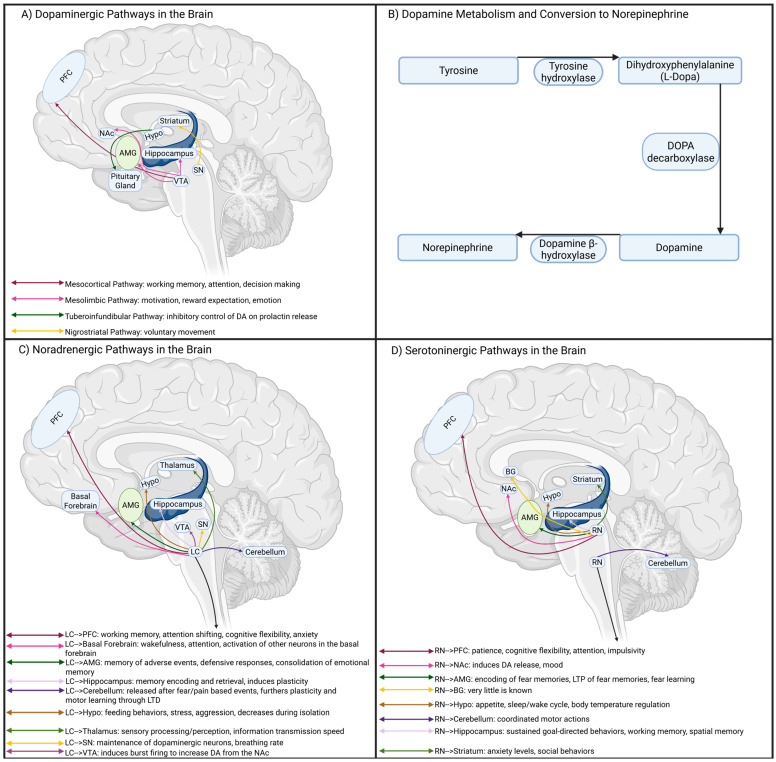
Monoamine pathways in the CNS. (**A**) Dopaminergic pathways within the CNS [83,84,85,86]. Mesocortical pathway emanates via projections from the VTA innervating the PFC to modulate concentration, focus, working memory, and decision-making. The mesolimbic pathway contains projections originating from the VTA that innervate the AMG, hippocampus, and NAc emerge to influence motivation, emotional processing, and reward-based decision-making. The tuberoinfundibular pathway involves innervations from DA neurons in the hypothalamus (hypo), which innervate neurons in the pituitary gland to provide negative feedback for the release of prolactin. Lastly, the nigrostriatal pathway emerges via projections between DA neurons in the SN to the striatum and is involved in regulating voluntary movement. (**B**) DA metabolism and conversion into NE [87,88,89]. (**C**) Noradrenergic signaling in the brain [90,91,92,93,94,95,96,97,98,99,100,101,102,103]. The LC is the primary site of NE synthesis from DA metabolism, and projects to the PFC alongside limbic structures of the brain. (**D**) Serotonergic signaling in the brain [104,105,106,107,108,109,110,111,112,113,114]. Notably, 5-HT transmission originates in the Rn and innervates limbic structures to modulate mood, hunger, and attention. Created in BioRender. Yacoub, M. (2025) https://BioRender.com/573y6zo.

**Figure 3 cells-14-01367-f003:**
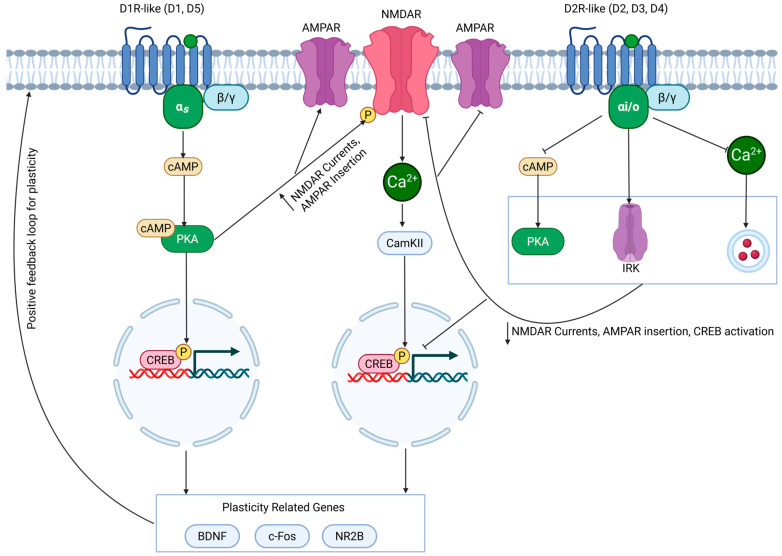
Interactions between NMDAR and DA signaling within the CNS. DA receptors are GPCRs classified as either D1R-like or D2R-like. Activation of DA receptors initiates a molecular cascade that utilizes proteins that overlap with NMDAR signaling. D2R-like activation inhibits cAMP production and therefore PKA signaling, alongside opening IRK channels and reducing calcium-mediated release of neurotransmitters. Altogether, this reduces the likelihood of neurotransmission occurring, and functions to inhibit NMDAR activity. D1R-like activation generally functions to increase the activation of PKA, which can lead to increased phosphorylation of NMDAR subunits and trafficking of AMPAR to synaptic sites. Additionally, PKA can phosphorylate the transcription factor CREB to further the transcription of genes involved in synaptic plasticity that underlie learning and memory. Created in BioRender. Yacoub, M. (2025) https://BioRender.com/xzfvvms.

**Figure 4 cells-14-01367-f004:**
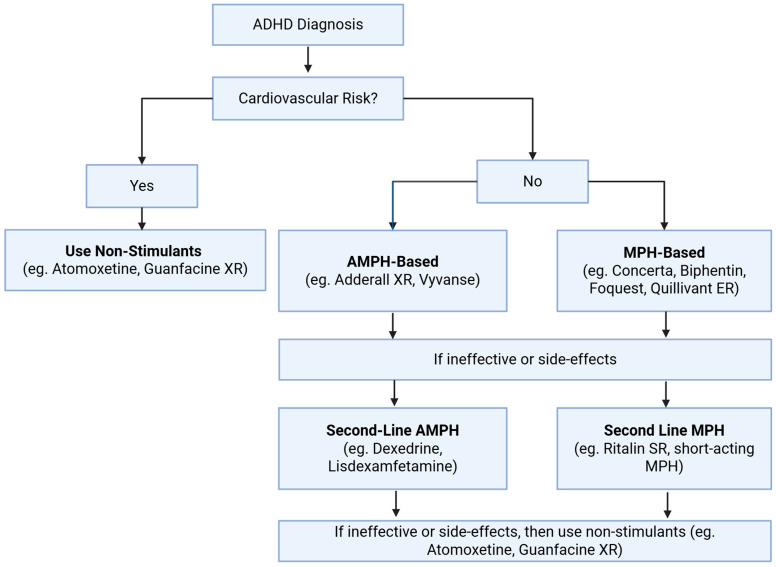
Overview of prescribing stimulants and non-stimulants for ADHD. Following an ADHD diagnosis, a family history for cardiovascular health is taken, alongside assessments of the patient’s heart health. If the patient does not present with risk for cardiovascular disease, the physician then considers a first-line treatment from either the AMPH- or MPH-based classes. In European countries, MPH-based medications are considered first; however, in North America, AMPH-based medications can also be considered. If the patient is a school-aged child, long-acting stimulants are preferred to ensure therapeutic benefits to the child throughout the entire school day. If first-line treatments prove ineffective, or present uncomfortable side effects, second-line treatments are then considered. Should these second-line treatments continue to produce undesired side effects, the patient may transition towards non-stimulants. If the patient does present cardiovascular risks following ADHD diagnosis, non-stimulants are prescribed instead. Created in BioRender. Yacoub, M. (2025) https://BioRender.com/krpeyqr.

**Figure 5 cells-14-01367-f005:**
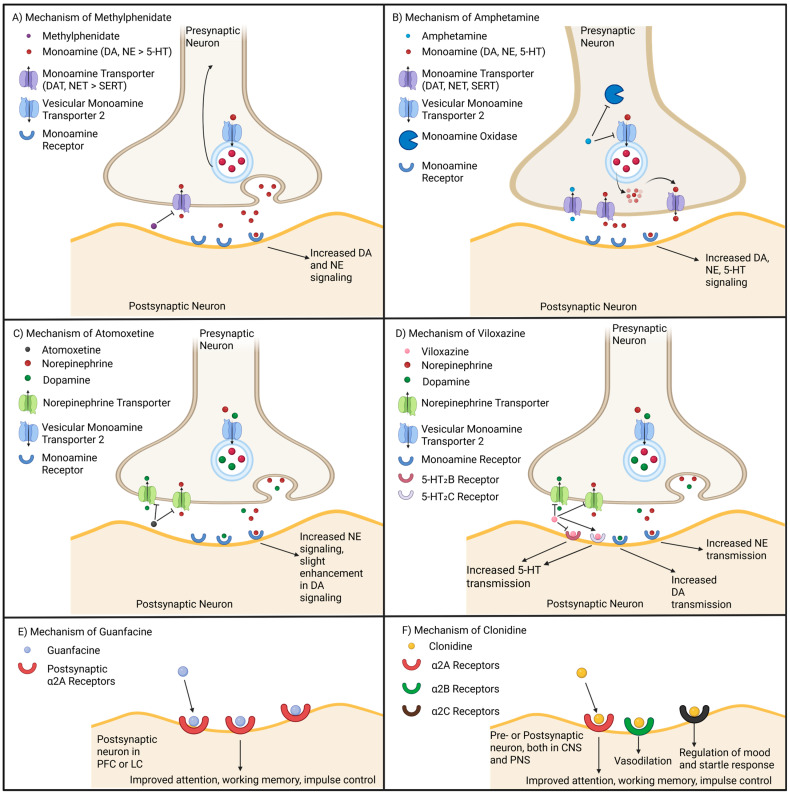
Molecular mechanisms and receptor targets of ADHD medications. (**A**) MPH inhibits DAT and NET but does not show strong affinity for blocking SERT, leading to increased DA and NE transmission. Additionally, MPH has been demonstrated to alter the distribution of vesicles in the presynaptic terminal, which is thought to allow for more controlled release of monoamines. (**B**) AMPH enters the presynaptic neuron using monoamine transporters and inhibits MAO to prevent monoamine breakdown. The inhibition of VMAT-2 prevents packaging of monoamines into vesicles, leading to the reverse transport of monoamines into the synaptic cleft to increase DA, NE, and 5-HT transmission. (**C**) ATX is a selective NET inhibitor, which causes an increase in the levels of NE available to bind to postsynaptic receptors. However, in the PFC these NETs also contribute to the reuptake of DA, thus ATX also increases DA transmission. (**D**) Viloxazine inhibits NET to increase NE signaling, alongside the availability of DA in the PFC. Additionally, viloxazine acts as an antagonist for 5-HT2B receptors, and an agonist for 5-HT2C receptors, which has been shown to increase 5-HT levels in the PFC. (**E**) Guanfacine binds to postsynaptic α2A receptors to inhibit cAMP activity, alongside PKA activation to reduce the opening of potassium channels. This enhances PFC neuronal activity to improve attention, working memory, and impulse control. (**F**) Similarly, Clonidine also binds to α2A receptors to exert similar effects, while binding to α2B and α2C receptors to increase vasodilation and regulation of mood and startle response, respectively. Created in BioRender. Yacoub, M. (2025) https://BioRender.com/z7ocirw.

**Figure 6 cells-14-01367-f006:**
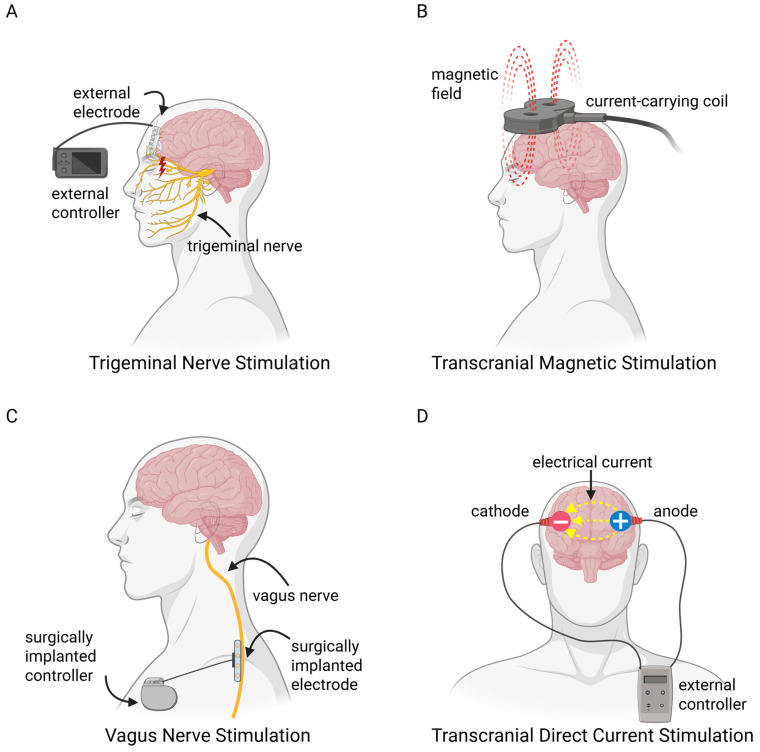
Modes of transcranial stimulation. (**A**) An external electrode is applied to the forehead of the patient to stimulate the trigeminal nerve. The electrode is controlled by an external controller. (**B**) A current-carrying coil is attached to the scalp which creates magnetic field by the coil interacting with the membrane potential of neurons. (**C**) An electrode is surgically implanted next to the vagus nerve to stimulate it with an electric current controlled by a pre-programmed, surgically implanted controller. (**D**) Two electrodes are attached to the scalp, and the current flows from the positively charged anode to the negatively charged cathode, which generates action potentials increasing neuronal firing in the brain. The electrodes are controlled by an external controller. Created in BioRender. Yacoub, M. (2025) https://BioRender.com/8or83ao.

**Table 1 cells-14-01367-t001:** Overview of ADHD subtypes and core symptom profiles.

Subtype	Diagnostic Criteria	Inattention Symptoms	Hyperactivity/Impulsivity Symptoms	Example Presentations
Primarily Inattentive (ADHD-I)	≥6 inattention symptoms (≥5 for adults); fewer than 6 hyperactivity/impulsivity symptoms (<5 for adults) [6]	“Often fails to give close attention to details or make careless mistakes in schoolwork, at work, or during other activities” [6] “Often has difficulty sustaining attention in tasks or play activities” [6] “Often do not seem to listen when spoken to directly” [6] “Often does not follow through on instructions and fails to finish schoolwork, chores, or duties in the workplace” [6] “Often has difficulty organizing tasks and activities” [6] “Often avoids, dislikes, or are reluctant to engage in tasks that require sustained mental effort” [6] “Often lose things necessary for tasks or activities” [6] “Is often easily distracted by extraneous stimuli” [6] “Is often forgetful in daily activities” [6]	Not meeting threshold [6]	Examples for Children: Frequently skipping entire test questions, or misreading class assignments [25]. Difficulty directing attention to class lectures, even if the lecture has just started [25]. Gets frequently distracted from completing tasks of chores to completion. A child with ADHD may start cleaning their room, but halfway through abruptly abandon the task to do something else [26]. Procrastinating completing homework or chores that will require sustained mental effort, even when they know the task is important [25]. Examples for Adults: Frequent employment changes, career instability, and challenges maintaining relationships [12]. Frequently making careless mistakes at work, such as sending emails with missing attachments, or overlooking or overlooking small details in reports [27]. Frequently misplacing key daily items such as car keys, cellphone, glasses, or work-related documents, even when aware that the items are required for the day [28].
Primarily Hyperactive (ADHD-H)	≥6 hyperactivity/impulsivity symptoms (≥5 for adults); fewer than 6 inattention symptoms (<5 for adults) [6]	Not meeting threshold [6]	“Often leave seat in situations when remaining in seat is expected” [6] “Often fidgets with or taps hands or feet or squirms in seat” [6] “Often runs about or climbs in situations where it is not appropriate (In adolescents or adults, may be limited to feeling restless)” [6] “Often unable to play or engage in leisure activities quietly” [6] “Is often ‘on the go’ as if ‘driven by a motor’” [6] “Often blurts out an answer before a question has been completed” [6] “Often talks excessively” [6] “Often has difficulty waiting his or her turn” [6] “Often interrupts or intrudes on others” [6]	Examples for Children: Repeatedly getting up during class to sharpen pencils or go to the bathroom due to difficulty remaining seated [29]. Often interrupting during peer conversations due to difficulty regulating conversational turn-taking, rather than intentional rudeness [8]. Examples for Adults: Internal or emotional restlessness: an adult with ADHD may feel that their mind or body cannot ‘settle’, even when they are sitting still [10]. Feels a strong urge to move, fidget, or shift positions, even while appearing outwardly relaxed [30]. Feeling urge to always complete a task, which may manifest in difficulty in enjoying relaxing endeavors [30]. In social interactions, an adult with ADHD may abruptly shift the topic of conversation before others have finished discussing the current subject [31].
Combined (ADHD-C)	≥6 inattention AND ≥ 6 hyperactivity/impulsivity symptoms (≥5 each for adults) [6]	Same 9 symptoms as ADHD-I [6]	Same 9 symptoms as ADHD-H [6]	Mixed profile of inattentive + hyperactive symptoms; may also include sensory processing difficulties, sleep difficulties [6].

Differences between the subtypes of ADHD: Primarily Inattentive (ADHD-I), Primarily Hyperactive (ADHD-H), and Combined (ADHD-C) using diagnostic criteria and symptoms as defined by the DSM-5 [6]. Diagnostic criteria require at least 6 symptoms (or 5 for individuals aged 17 and older) within each domain for ADHD-C. Diagnosis is characterized by several symptoms prior to age 12 years and those occurring in ≥2 settings and impede “normal” function or development. For diagnosis, these symptoms are not better explained by another mental disorder.

**Table 2 cells-14-01367-t002:** Gene variants associated with ADHD development and symptoms.

Candidate Gene	Basal Function	Gene Variation/Single Nucleotide Polymorphism	Clinical Phenotype	References
*SLC6A2/NET*	Encodes for the NE transporter which regulates NE homeostasis.	rs36011	Deficits in visual memory and attention	[134,156]
		rs1566652		
		rs11568324	Implicated in ADHD heritability and etiology	[157]
*SLC6A3/DAT*	Human DA transporter gene; regulates DA reuptake and neurotransmission dynamics	10-repeat (10R) allele in 3′ UTR	Homozygosity implicated with worse response inhibition and sustained attention	[158,159,160]
*SLC6A4/SERT*	5-HT transporter gene associated with socio-emotional development and stress susceptibility.	Promoter methylation	Associated with more hyperactive-impulsive symptoms, and reduced cortical thickness in occipito-temporal regions	[161,162]
*DBH*	The protein encoded by this gene converts DA to NE.	*TaqI* polymorphism (T)	Poorer sustained and visual attention	[138,139,140]
*ADGRL3 (LPHN3)*	Encodes for protein ADGRL3 and signals for cell-cell adhesion, signal transduction, and contributes to CNS development.	rs35106420	Variants associated with increased susceptibility to developing ADHD	[190,191,192]
		rs61747658		
		rs734644		
*DRD4*	Encodes for highly expressed DA receptor 4, regulates the dopaminergic pathway, and is involved in perception and response to sensory stimuli and executive function.	Exon III 48 base pair variable number tandem repeats	Effects on functional brain connectivity, deficiencies in perceptual organization, and attention symptoms	[167,168,169]
		rs916457		
*DRD1*	Encodes DA receptor 1 and plays a role in attention.	rs265977	Linked to hyperactivity and impulsivity symptoms	[164,165]
*COMT*	Enzyme product of gene catabolizes extra-neuronal DA and is linked to cognition, attention, and social behavior.	rs4680	Linked to inattentive and hyperactive symptoms, as well as decreased grey matter volume	[193,194,195]
*ADRA2A*	Encodes for the alpha-2A-adrenergic receptor which is linked to working memory and focused attention through noradrenergic projections to the PFC.	rs553668	Gene polymorphisms have been linked to severity of inattentive symptoms, and impacts on auditory/visual attention	[172,173]
		*MspI* site in the promoter region (C→G) SNP		
*MAOA*	The protein product of the gene is an enzyme that affects metabolism of dopamine to noradrenaline.	rs5906883	Gene polymorphisms potentially alter catecholamine regulation and are correlated with increased susceptibility to ADHD development and symptoms	[163]
		rs3027407		
*CDH13*	Encodes for an atypical cadherin that drives neurodevelopmental processes and may have multiple functions related to signaling, neurite growth, and synapse development.	rs2199430	Linked with excitatory/inhibitory imbalance, hyperactive-impulsive symptoms, deficits in learning and memory, and lower agreeableness	[178,179,180]
		rs11150556		
*CTNNA2*	Encodes an alpha-catenin protein involved in neuronal migration, synaptic plasticity, and is critical for synaptic contact.	rs7600563	Variants are associated with impulsivity, excitement seeking	[181,182,183]
		rs13395022		
*DRD2*	Encodes for D2 receptors.	rs2075654	Greater rate of commission errors.	[166]
		rs1079596		
*DRD5*	Encodes for D5 receptors which stimulates adenyl cyclase activity.	Dinucleotide repeat of 148-bp allele (risk factor) is located 18.5 kb from 5′ end.	148-bp allele is a risk factor for ADHD, and also significantly predicts baseline persistent ADHD diagnosis. 136-bp allele is a protective factor.	[170,171]
		136-bp allele was a protective factor.		
*SNAP-25*	Encodes for the SNAP-25 protein, which is involved in the SNARE complex to facilitate synaptic vesicle fusion and neurotransmitter release.	rs3746544 located in the 3′ UTR	Association with ADHD.	[196]
*PPP1R16A*	Encodes a myosin phosphatase targeting subunit, which guides catalytic protein phosphatase 1 subunit to its intended targets to facilitate control of the actin cytoskeleton and cell-cell junctions.	Chromosome 1	Reduced expression in ADHD cases, linked to educational attainment	[185]
*B4GALT2*	Encodes for an enzyme involved in the synthesis of glycoconjugates and saccharide structures.	Chromosome 8	Reduced expression in ADHD cases, linked to educational attainment.	[185]
*SORCS3*	Encodes for the SORC3 protein which is an intracellular trafficking receptor for tropomyosin-related kinase B. Highly expressed in the CA1 region of the hippocampus and localized to the postsynaptic region. Regulates postsynaptic depression and influences the extinguishing of fear-related memories.	Deleterious variants	Linked to increased risk of ADHD.	[184,185,186]
*NR3C1*	Encodes for the glucocorticoid receptor, which regulates processes including stress response, metabolism, immune regulation, mood, and cognition.	Haplotype TthIIII-NR3C1-I-ER22/23EK	Potentially linked to increased risk of ADHD.	[197,198]
*EPHA5*	Encodes for the EphA5 receptor which plays a role in synaptic plasticity. Interactions between ephrin-A5 and the EphA5 receptor induces NMDAR-PSD95 complexes, spine maturation, and neuronal activity during early hippocampal development.	rs4860671	Reduced performance in working memory and processing speed.	[199,200,201]
*CDH13*	Encodes for an atypical cadherin that drives neurodevelopmental processes, and may have multiple functions related to signaling, neurite growth, and synapse development.	rs8055161	Linked with excitatory/inhibitory imbalance, hyperacuity/impulsivity symptoms, deficits in learning and memory, and lower agreeableness	[178,179,180]
		rs6565113		
		rs11150556		
		rs2199430		
*LIME1*	Encodes a transmembrane adaptor protein and plays role in inflammatory signaling pathway.	Increased methylation at:	Increased attention deficits as measured by Conner’s Continuous Performance Test	[202]
		cg00446123		
		cg20513976		
*SPTBN2*	Regulates glutamate signaling pathway and is associated with neurodevelopment.	Reduced methylation at cg02506324	Increased attention deficits as measured by Conner’s Continuous Performance Test	[202]
*HTR1B*	Encodes for the 5-HT 1B receptor (HTR1B).	rs6296	Controls the release of DA, 5-HT, and acetylcholine in the brain. Knockout mice demonstrate behavioral disinhibition, hyperactivity, and increased aggressiveness.	[174,175,176,177]
		rs6297		
		rs11568817		
		rs130058		
		rs6298		
		rs130060		

**Table 3 cells-14-01367-t003:** Overview of Commonly Used ADHD Animal Models. Summary of ADHD animal models, including their method of generation or manipulation, ADHD-relevant behavioral phenotypes, biological alterations, advantages, and limitations.

ADHD Animal Model	Manipulation	ADHD Relevant Features	Behavioral Phenotypes	Biological Alterations	Advantages	Disadvantages	References
DAT mutant mice	DAT KnockOut/Down	Hyperactivity	Excessive locomotion.	Fivefold-higher extracellular DA concentrations, 300-fold reduced DA reuptake.	Achieves strong hyperactivity phenotype. Useful for investigating molecular mechanisms linking DA signaling and hyperactivity.	Knockout of DAT prevents AMPH and MPH from exerting one of their core mechanisms to improve ADHD symptoms. Reduced DA is not a universal hallmark of human ADHD. Complete knockout of DAT may lead to developmental compensations that limit translational value.	[230,231,235,236,237]
Spontaneously Hypertensive Rat Model	Crossing male Wistar rats with spontaneous hypertension with females with moderately high blood pressure. Mate hypertensive rats again.	Attention deficit, hyperactivity, impulsiveness. Psychostimulants improve core ADHD-like symptoms in model.	More sensitive to delays in reinforcement similar to ADHD children.	Increased D1/D5 receptor density in neostriatum and NAc for young males. Reduced D4 gene and protein expression in PFC. Increased NE production and release in PFC.	Behavioral symptoms resemble core ADHD symptoms. Responds to stimulants like AMPH and MPH.	Confounding influences of hypertension on ADHD symptoms may reduce translatability of findings to human ADHD. Most studies use Wistar-Kyoto rats as controls, which have increased depression behaviors.	[231,238,239,240]
Coloboma Mouse	Generated through an irradiated deletion of 2cM region on chromosome 2.	Hyperactivity, impulsivity, inattention Are somewhat responsive to AMPH treatment, but not MPH.	Reduced patience in delayed reinforcement task. Increased locomotor activity.	Reduced DA release from dorsal striatum. Increased TH mRNA alongside increased α2A-receptor mRNA in LC. DA levels are increased in ventral striatum and cerebral cortex.	SNAP-25 variants are linked to ADHD in humans. Do show ADHD-like symptoms that are somewhat treated by stimulants.	Lethality in homozygotes, only heterozygotes. Generation of strain involves deletion of other genes beyond SNAP25, leading to confounding causes of the phenotype.	[231,241,242,243,244,245,246]
α-synuclein-lacking mice	Knocking out α-synuclein.	Hyperactive	Memory deficits	Elevated DA	Impacts DA function, allowing for investigating how DA impacts ADHD symptoms.	Does not strongly mimic inattention associated with ADHD. Reduced effect of D-AMPH as compared to WT mice.	[239,247]
Neonatal 6-hydroxydopamne Lesion Rat	Neonatal rats are injected with 6-hydroxydopamine which targets developing dopaminergic neurons to yield DA depletion in striatal/cortical targets.	Hyperactivity, attention-deficit, and impulsiveness. MPH reduced hyperactivity symptoms.	Adolescent mice exhibit anxiety, and antisocial behavior.	Reduced TH in the striatum, SN, and VTA.	Targets the DA system which is implicated in ADHD. Lesions in the neonatal model produce a neurodevelopmental origin of ADHD rather than adult-onset pathology. Allows for relatively selective DA manipulation.	More invasive as compared to SHR or genetic models. Exposure to DA neurotoxin during early development may result in compensatory changes.	[248,249]

## Data Availability

No new data was created or analyzed in this study. Data sharing is not applicable to this article.

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
