# Peer review of "Attention-Deficit Hyperactivity Disorder (ADHD): A Comprehensive Overview of the Mechanistic Insights from Human Studies to Animal Models"

_cells, 2025, doi:10.3390/cells14171367_

Round 1

Reviewer 1 Report

Comments and Suggestions for Authors

The paper entitled "Attention Deficit Hyperactivity Disorder (ADHD): A Comprehensive Overview of the Mechanistic Insights from Human Studies to Animal Models" is well-written and has the potential to be published in the Cells journal. In this review, the authors define ADHD as a complex neurodevelopmental disorder with distinct trajectories across the lifespan. While progress has been made in identifying behavioral features and treatment strategies, significant gaps remain in the understanding of developmental mechanisms underlying the condition. Clarifying these mechanisms is critical for improving diagnostic accuracy in adults, assessing the safety of long-term pharmacological interventions, and evaluating the potential impact of prenatal exposure to ADHD medications on the developing brain. I have major criticisms of this work.

Major comments: 1) This review is missing important information on the genes involved in ADHD development. The authors should include this information in a table format for the reader’s convenience.

2)The authors are encouraged to present the animal models used in ADHD research in a table format for greater clarity and ease of reference.

Minor: Figure 1 is not appropriate for a scientific paper. The authors should improve it to make it look more professional, or alternatively, present this information in a table format.

Reviewer 2 Report

Comments and Suggestions for Authors

This is a comprehensive, well-written and stimulating review of Attention Deficit Hyperactive Disorder (ADHD). Given the prevalence of this "disorder", the concerns about over-medication of school age children simply to provide quieter classrooms, medication side effects and potential toxic effects of ADHD medications upon fetal and childhood brain development, this review could not be more timely. Overall, the authors do an excellent job of avoiding biases and present ADHD in both children and adults as neurodevelopmental syndrome that is heterogenous both in symptoms and responses to medications. 

In site of its length, I enjoyed reading this review. There are >300 references, most are from 21st century work and many are from the last 5 years. Thus, this review is contemporary. The English is fine, and the only suggestion I have is for the authors to increase font size in many of the otherwise excellent figures. There is also a paucity of discussion of recent molecular genetic and some important brain functional imaging studies of ADHD subjects. 

As the authors appreciate (and discuss), ADHD today is still mostly a behavioral syndrome diagnosis that can overlap with other (potentially over-diagnosed) behavioral syndromes such as Autism Spectrum Disorder (ASD). In spite of the authors' presentation of contemporary concepts about ADHD neurobiology/neuropharmacology, there are as of yet no "biomarkers" that are helpful in either diagnosis or evaluating responses to medications. That will have to wait for future research, and the authors discuss many areas where research is needed. Perhaps functional brain imaging will help in this area, and the authors provide clues (particularly in Section 3) that this is the case.

I felt that the Figures were educational and appropriate. Again, my only major concern is that the font sizes are too small.

Round 2

Reviewer 1 Report

Comments and Suggestions for Authors

I recommend accepting this paper in its present form.